# Identification of CCZ1 as an essential lysosomal trafficking regulator in Marburg and Ebola virus infections

Vanessa Monteil [1], Hyesoo Kwon [1,2,17], Lijo John [2,17], Cristiano Salata [3,17], Gustav Jonsson [4,5], Sabine U. Vorrink[6], Sofia Appelberg [7], Sonia Youhanna[6], Matheus Dyczynski[8,9], Alexandra Leopoldi[4], Nicole Leeb[4], Jennifer Volz[4], Astrid Hagelkruys [4], Max J. Kellner[4,5], Stéphanie Devignot [1], Georg Michlits [8,9], Michelle Foong-Sobis[4], Friedemann Weber [10], Volker M. Lauschke [6,11,12], Moritz Horn[8,9], Heinz Feldmann [13], Ulrich Elling [4], Josef M. Penninger [4,14,15,16] & Ali Mirazimi [1,2,7] ✉

Marburg and Ebola filoviruses are two of the deadliest infectious agents and several outbreaks have occurred in the last decades. Although several receptors and co-receptors have been reported for Ebola virus, key host factors remain to be elucidated. In this study, using a haploid cell screening platform, we identify the guanine nucleotide exchange factor CCZ1 as a key host factor in the early stage of filovirus replication. The critical role of CCZ1 for filovirus infections is validated in 3D primary human hepatocyte cultures and human blood-vessel organoids, both critical target sites for Ebola and Marburg virus tropism. Mechanistically, CCZ1 controls early to late endosomal trafficking of these viruses. In addition, we report that CCZ1 has a role in the endosomal trafficking of endocytosis-dependent SARS-CoV-2 infections, but not in infections by Lassa virus, which enters endo-lysosomal trafficking at the late endosome stage. Thus, we have identified an essential host pathway for filovirus infections in cell lines and engineered human target tissues. Inhibition of CCZ1 nearly completely abolishes Marburg and Ebola infections. Thus, targeting CCZ1 could potentially serve as a promising drug target for controlling infections caused by various viruses, such as SARS-CoV-2, Marburg, and Ebola.

The *Filoviridae* family comprises viruses with filamentous enveloped virions containing a non-segmented negative-strand RNA genome. Several viruses of this family, such as Ebola virus (EBOV) and Marburg virus (MARV), are known to cause hemorrhagic fever in humans and non-human primates. Despite great efforts to develop effective therapeutic strategies, thus far only two vaccines[1] (a single shot vesicular stomatitis virus [VSV]-EBOV vaccine and the Ad26.ZEBOV/MVA-BN-Filo prime-boost) and two monoclonal antibody-based treatments (https://www.cdc.gov/vhf/ebola/treatment/index.html) have been approved against Ebola virus infections. Thus, it is paramount to identify novel

and, critically, universal pathways that are essential for Ebola and Marburg infections.

Filoviruses enter host cells mainly through macropinocytosis and subsequently are trafficked from early to late endosomes/lysosomes, where the proteolytic processing of glycoproteins (GP) takes place, mediated by cathepsins and related host proteases[2–4]. Genome-wide screening methods have facilitated and accelerated the identification of host genes that play a role in infections. Among these systems, haploid cells offer genetic screening capabilities to identify host factors that might be otherwise masked. The main advantage of haploid

cells compared to diploid cells is the unequivocal presentation of recessive phenotypes as cells only contain a single genomic copy. Indeed, haploid screens have been successfully used to uncover host dependency factors for defined viruses[5–11]. To date, several proteins involved in Ebola virus infections have been identified such as the Niemann-Pick C1 protein (NPC1), the vacuole protein sortin complex cathepsin B (CTSB), the lipid kinase PIKFYVE16 and the N-acetylglucosamine-1-phosphate transferase alpha and beta subunits (GNPTAB)[12,13].

To uncover additional host factors for filovirus infections, we used our haploid murine stem cells (mSC) mutagenized with revertible transposon and viral-based gene-trap vectors[14]. Using these tools, we identified CCZ1, a guanine nucleotide exchange factor involved in endolysosomal trafficking, as an essential host factor for Marburg and Ebola virus infections but not infection of the Lassa arenavirus (LASV). Moreover, CCZ1 is involved in endocytosis-dependent SARS-CoV-2 infections.

## Results

### Haploid stem cells screening uncovers host factors for Marburg virus infections

Various screening strategies have been used to identify the critical factors involved in filovirus infection. Here, we utilized genome-wide screening in murine haploid embryonic stem (mES) cells to identify the potential host reliance factors in Marburg virus infection. We initially subjected mES cells to mutagenesis using revertible transposon and viral-based gene-trap vectors, (insertional mutagenesis)[14]. This resulted in a diverse collection of cells with gene disruptions. One advantageous feature of this system is that each gene-trap construct is equipped with a unique barcode. By performing integration PCR and sequencing this fragment, we can identify the specific gene that has been targeted, eliminating the need for extensive sequencing of the entire cell DNA. Subsequently, we infected the cells with a lytic virus. As a consequence, cells lacking non-essential genes for viral infection succumbed to the infection and died. On the other hand, cells carrying disruptions in genes that are crucial for the viral infection cycle survived due to their resistance to cell death caused by the virus. Considering the technical constrains due to the use of a class 4 agent, we

adopted a replication competent vesicular stomatitis virus pseudo-typed with the Marburg virus glycoprotein (VSVΔG/MARVGP) (Fig. 1a)[15]. In addition to the reduced biosafety requirements, VSV is an excellent screening system because these pseudotyped virus infections results in near 100% cell death. Indeed, the VSVΔG/MARVGP efficiently replicated in and killed mouse haploid AN3–12 cells (Fig. 1b), allowing the selection of the mutagenized AN3–12 cell that survived to VSVΔG/MARVGP infection. The process is described in Fig. 1c, indicated that in these cells, gene(s) knocked out are essential for the virus infection step.

Using this experimental workflow, we infected 100 million haploid cells with VSVΔG/MARVGP and the VSVΔG/MARVGP resistant cells were subsequently expanded. Among the resistant cells, the site of the gene-trap insertions was mapped using deep sequencing. Using this setup, we identified multiple genes involved in the endo/lysosomal network (*Npc1*, *Ccz1*, *3110002H16Rik*, *Rab7*, *Rab12*, *Rmc1*, *Ctsl*), Broad spectrum antiviral factors (*Ifitm1*, *Ifitm2*), components of the ER membrane protein insertion complex (*Emc8*, *Emc2*, *Ssr2* and the *RP24-414A22,6* gene with unknown function). Other hits identified are involved in the regulation of cell growth and proliferation (*Klf5*, *Etl4*, *Ppp2r5c*) (Fig. 2a–c). The top hit was NPC1, confirming a previous screen for filoviruses[13]. Among the genes with the highest number of unique gene-trap insertions are two known Filovirus restriction factors of the IFITM protein family, *Ifitm1* and *Ifitm2*[16,17]. Genes belonging to the endo/lysosomal network were strongly enriched in screen hits, with *Ccz1* and *Rmc1* showing the most gene-disrupting insertions among those (Fig. 2b). CCZ1 is part of the CCZ1-MON1-RMC1 complex that is indispensable for endocytic trafficking (Fig. 2c). We therefore focused on CCZ1, which has never been shown to be involved in filovirus infections. To understand the specificity of our screen to identify Filovirus entry host factors, we performed a similar haploid cell screen with rVSV expressing LASV glycoproteins (VSVΔG/LASVGPC). While this screen revealed several genes known to be important for LASV entry (*B3galnt2* and *Dag1*), we could not find any members of the CCZ1-MON1 complex enriched among significant hits[18] (Supplementary Fig. 1). Thus, our screening system using VSVΔG/MARVGP and, as a control, VSVΔG/LASVGPC, identified genes previously shown to be implicated in EBOV, MARV and LASV

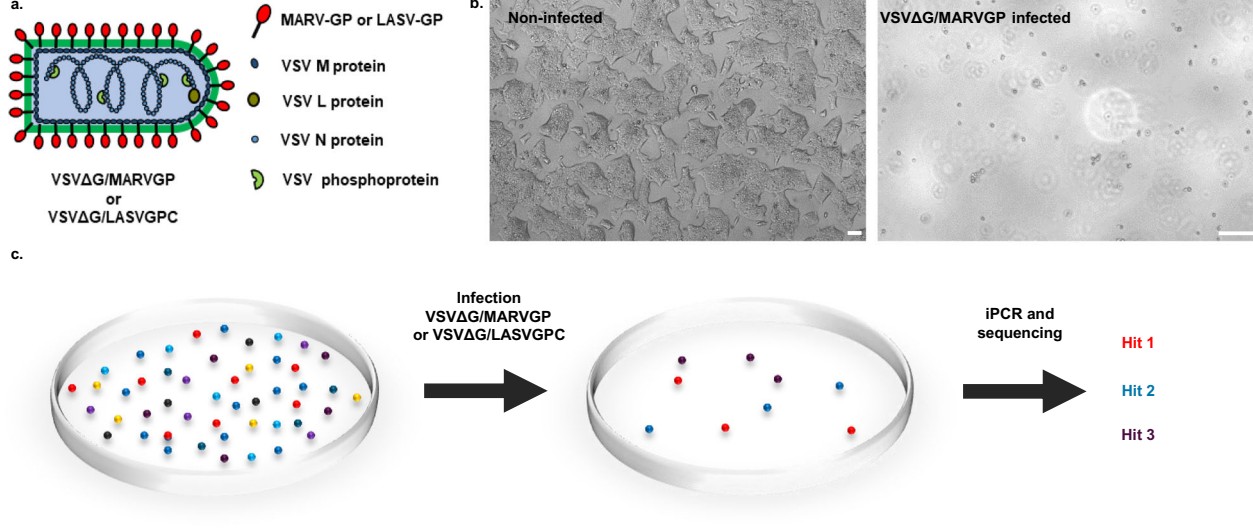

**Fig. 1 | Principle of the haploid cells screening. a** Schematic drawing of the recombinant VSV-GP viruses. **b** Microscopy of AN3–12 cells non-infected (left) or infected VSVΔG/MARVGP (right) 72 h post infection at a multiplicity of infection (MOI) of 1. Scale bar 100 μm. Pictures are representative of 3 independent experiments. **c** Schematic drawing of the haploid cells screening steps. The library

of AN3–12 cells mutagenized by insertional mutagenesis was infected with VSVΔG/MARVGP or VSVΔG/LASVGPC. The surviving cells were pooled and their genomic DNA was analyzed by integration PCR (iPCR) and sequencing to find the disrupted genes.

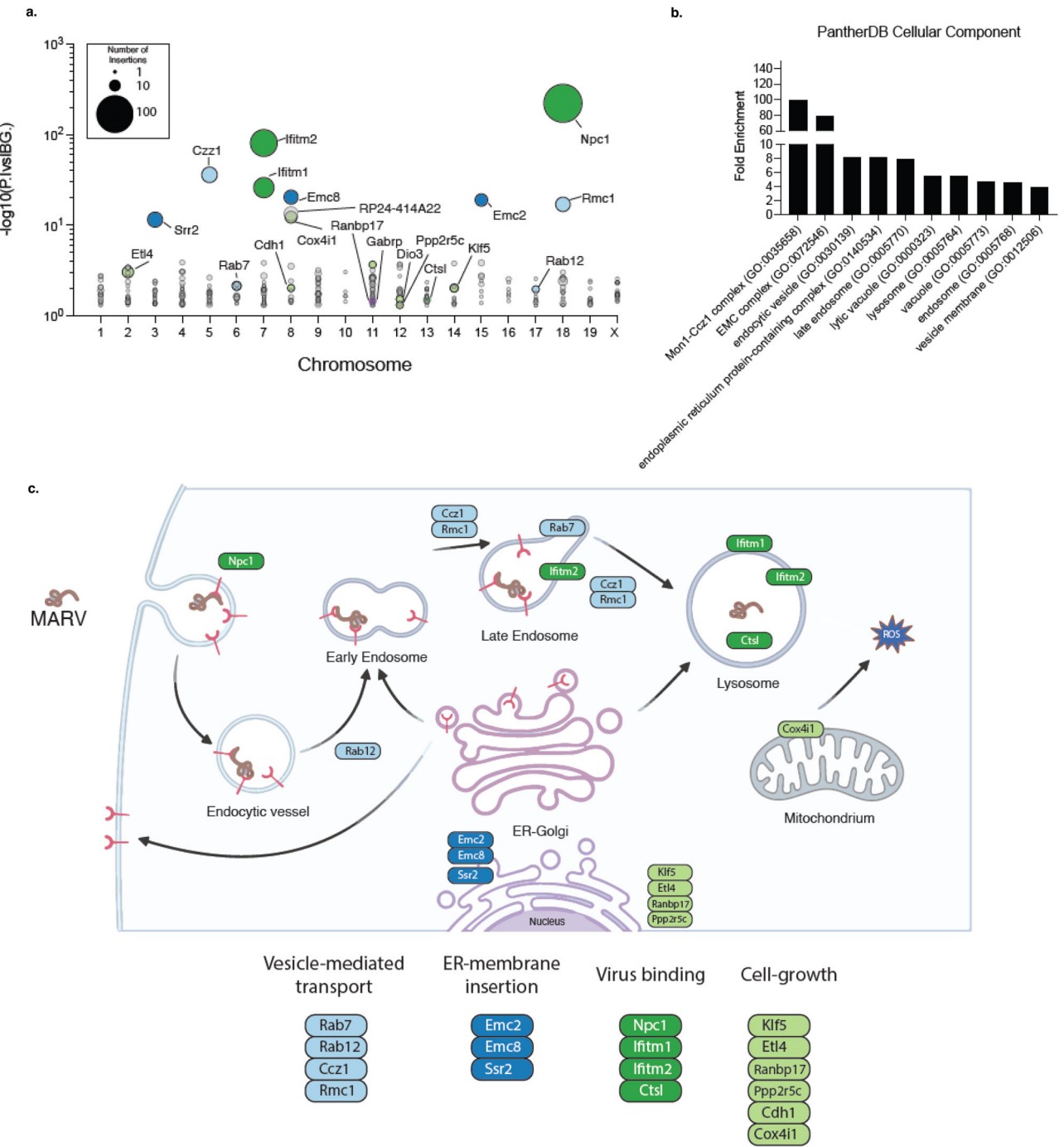

**Fig. 2 | Identification of VSVΔG/MARVGP genes of resistance. a** Bubble-plot showing hits identified from the VSVΔG/MARVGP gene-trap insertion screen. Hits are shown as individual dots, grouped by chromosomal location and stratified along the y-axis by statistical significance (two-tailed binomial test, no adjustment) of gene insertions found in the screen over background. Dot size indicated the number of unique insertions found. Genes with more than four unique insertions are color-coded by functional groups and labeled. Raw data are presented in Source data. **b** Gene Ontology (GO) enrichment analysis for cellular component is shown. Displayed are the Top 10 most enriched terms, ordered by fold enrichment derived from the PantherDB analysis using significant hits identified in the screen. **c** A schematic illustrating the putative involvement of significant hits identified in the gene-trap mutagenesis screen. Below are the assigned functional categories. The scheme was drafted using Biorender (Biorender.com) and further modified with Adobe Illustrator.

infections and, importantly, uncovered candidate host genes for MARV infections.

### CCZ1 as an important entry factor for filoviruses in haploid cells

To confirm that CCZ1 has a role in filovirus entry, wild type and *ccz1* knocked out AN3−12 cells by insertional mutagenesis were infected with VSVΔG/MARVGP and assessed for relative vRNA titer by qRT-PCR. Insertional mutagenesis allowing the reversion of the gene-trap

leading to the reversion of the knockout, this inverted sister clone was used as a control. To validate the system, as both RAB7 and NPC1 proteins have been previously reported as important factors for filovirus infections[6,13,19], *rab7* and *npc1* knocked out cells as well as their respective inverted sister clones were also infected with VSVΔG/MARVGP. All cells were produced and validated by Haplobank (IMBA). As expected, the knockout of *rab7* and *npc1* lead to a dramatic decrease in infection (98.4% and 99.99% respectively), mainly

recovered in inverted cells (Fig. 3). More importantly, the knockout of *ccz1* also lead to a dramatic drop (98.9%) in virus infection which is also mainly recovered in inverted cells (Fig. 3). These data confirm the importance of CCZ1 in virus entry in haploid cells.

**CCZ1 as an important entry factor for filoviruses in diploid cell**
To confirm our hits, we generated human diploid A549 knockout (KO) cells for either *CCZ1*, *RAB7* or *NPC1* genes using CRISPR/Cas9 technology and cells were controlled for knockout by western blot

(Supplementary Fig. S2). Of note, Wild-type (WT) cells, control cells (cells subjected to CRISPR/Cas9 treatment using scrambled RNA guides) as well as 3 different KO clones for each of these genes were infected with rVSV-GP-MARV and cells were analyzed by western blot. Deletions of *NPC1*, *CCZ1* and *RAB7* lead to a marked decrease in VSVΔG/MARVGP infection as determined by a decrease in VSV-M protein (Fig. 4a–c). For each gene, the clone showing the strongest resistance to VSVΔG/MARVGP infections (*CCZ1* clone 2, *NPC1* clone 2 and *RAB7* clone 3) was selected. GFP-expressing WT cells were then mixed with

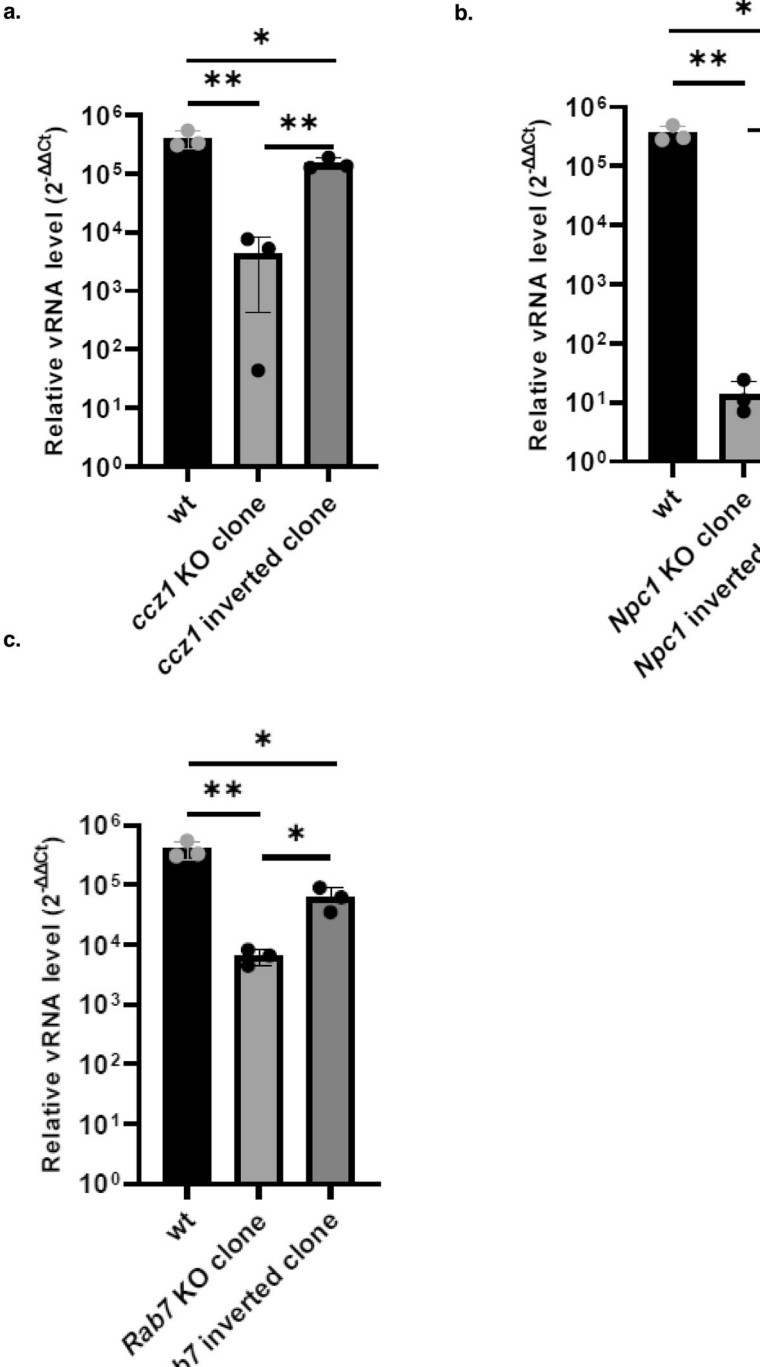

**Fig. 3 | Validation of major hits using VSVΔG/MARVGP in haploid cells.** Wild-type, *ccz1* (**a**), *npc1* (**b**) and *rab7* (**c**) knocked out haploid cells as well as their inverted sister clones (Genetrap insert reverted to invert the knockout) were infected with VSVΔG/MARVGP and assessed for relative level of infection to wild-type cells (qRT-PCR of MARV RNA/endogenous control ($2^{-\Delta\Delta Ct}$) compare to wild-type cells). Reverting the knockout lead to a reversion of the deleterious effect of the knockout for all genes. Error bar represent mean ± SD, $n = 3$ biologically independent replicates. *$P < 0.05$, **$P < 0.01$. Two-tailed Student's *t*-test. Exact $p$ values and Source data are provided as a Source Data file.

these three (GFP-negative) clones and were either mock infected or infected with VSVΔG/MARVGP for 48, 72 and 120 h. At each time point, the percentages of GFP-expressing WT cells and GFP-negative KO cells was determined by flow cytometry. Gating strategy is presented in Supplementary Fig. 3. Knockout cells for *CCZ1*, *NPC1* or *RAB7* genes survived the VSVΔG/MARVGP infection and expanded over time, in contrast to GFP-expressing WT cells (Fig. 4d), confirming the role of these genes in VSVΔG/MARVGP infections. As a control, *CCZ1* knocked out cells (clone 2) were transcomplemented using an expression plasmid coding for *CCZ1*. These cells were infected with VSVΔG/MARVGP for 48 h and and assessed for relative vRNA titer by qRT-PCR. The transcomplementation of *CCZ1* lead to a partial reversion of the deleterious effect of *CCZ1* knockout (Fig. 4e).

Because CCZ1 is in complex with MON1 (itself formed by MON1A and MON1B), we knocked out MON1A and MON1B respectively and assessed for infection with VSVΔG/MARVGP. The knockout of any of these proteins lead to an important drop in infection (89.4% for MON1A and 82.4% for MON1B), also partially recovered when the cells are transcomplemented (Fig. 4f, g).

All these results highlight the role of CCZ1 in filovirus endocytosis, while in complex with MON1.

To confirm these data in a real infection system, A549 cells were infected with MARV at a MOI of 0.1 for 48 h. As shown in Fig. 5a, knocking-out *CCZ1* and *RAB7* resulted in more than 98% decrease in the relative level of infection, whereas we observed residual infection in *NPC1* mutant A549 cells. Thus, the knockout of *CCZ1* and *RAB7* in particular, confers near complete resistance to MARV infections in human cells. Immunostaining analysis of *CCZ1* KO cells showed a 80% decrease in the number of cells infected with VSVΔG/MARVGP and more than 99% decrease in the number of cells infected with MARV

(Fig. 5b). Picture of infected cells are presented in Supplementary Fig. 4. To confirm the role of our hit CCZ1 for filovirus infections in general, we infected *CCZ1* mutant A549 (clone 2) as well as WT cells and control cells (cells subjected to CRISPR/Cas9 treatment using scrambled RNA guides) at a MOI of 0.1 with MARV, EBOV and, as a control, LASV isolates. As shown in Fig. 5c, the knockout of *CCZ1* had no effect on LASV infections but resulted in a significant decrease in EBOV and MARV infections. These genetic data identify CCZ1 as a critical host factor for Marburg and Ebola virus infections.

## CCZ1 knockdown suppresses MARV infection in 3D organotypic primary human hepatocyte culture

Hepatocytes are one of the key cellular targets for filoviruses. Primary human hepatocyte (PHH) spheroids are suitable culture systems that closely resemble the in vivo human liver at the transcriptomic, proteomic and metabolic level[20–22]. We therefore subjected 3D liver spheroids to infections with VSVΔG/MARVGP. To validate the role of *CCZ1*, PHH were transfected with a siRNA targeting *CCZ1* and knockdown of the gene was verified by western blotting 3 days post seeding (Fig. 6a). The densitometry analysis of the western blot show that the knockdown of *CCZ1* by siRNA lead to a 52,6% decrease in CCZ1 protein expression compare to non-transfected cells (Fig. 6a). Hepatocytes in suspension and liver spheroids were infected with VSVΔG/MARVGP. Notably, knockdown of *CCZ1* resulted in significant reductions of viral infection (75.2% for PHH suspension MOI 1, 90.8% for PHH suspension MOI 5, 68.5% for PHH spheroid MOI 1 and 74.1% for PHH spheroid MOI 5), irrespective of infection time, as monitored by viral-specific qRT-PCR (Fig. 6b). Thus, VSVΔG/MARVGP can efficiently infect human hepatocytes in suspension or in 3D liver spheroids and the knockdown of *CCZ1* reduces infectivity.

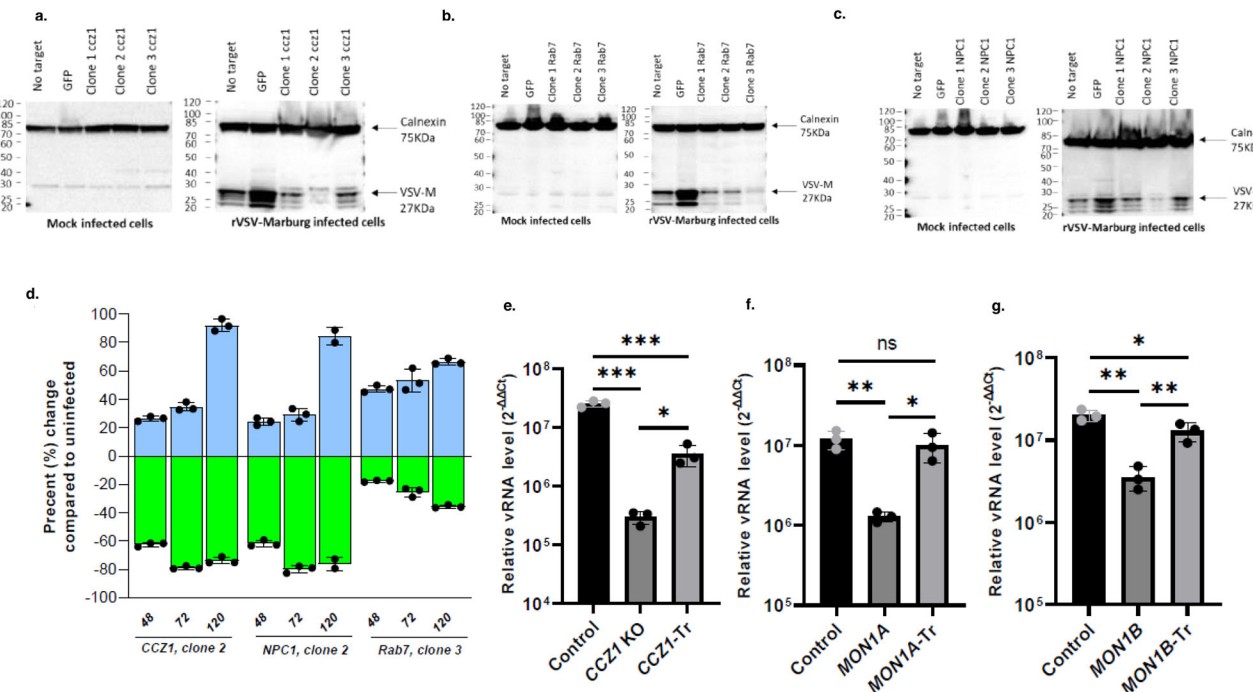

**Fig. 4 | Validation of major hits in diploid cells using VSVΔG/MARVGP.** Western blot analysis of the levels of VSV-M protein in various clones of A549 KO cells targeting CCZ1 (**a**), RAB7 (**b**) and NPC1 (**c**), 48 h post infection with rVSV-GP-MARV. Western blot showing the KO of CCZ1, NPC1 and Rab7 are shown in Source data linked to this manuscript. **d** Survival ratio of mutant (unstained) to WT (GFP) A549 after infection with VSVΔG/MARVGP. The bar graph is representative of 3 replicates. The bar graphs show the change in percentage compared to mixed, uninfected population at each time point. The green bars are on a negative y-axis representing a growing decrease in GFP+ cells (WT cells) in the mixed, infected population compared to mixed non-infected population. The blue bars on the other hand, is on a positive y-axis showing a growing increase in GFP- (KO cells) cells in the mixed, infected population compared to mixed non-infected population. Gating strategy is presented in Supplementary Fig. 3. Relative level of infection of *CCZ1* KO (**e**), *MON1A* KO (**f**), *MON1B* KO (**g**) and respective transcomplemented (Tr) cells to control cells infected with VSVΔG/MARVGP. Error bars represent mean ± SD, *n* = 3 biologically independent replicates. *$P < 0.05$, **$P < 0.01$, ***$P < 0.001$; two-tailed Student's *t*-test. Exact *p* values and Source data are provided as a Source Data file.

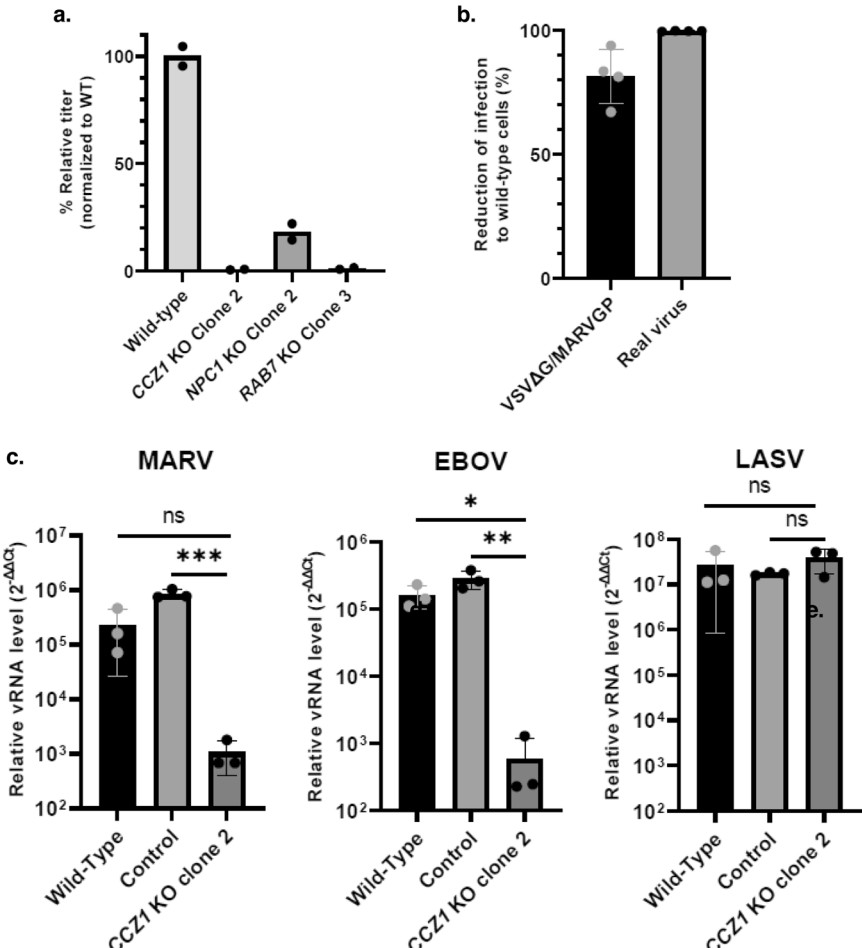

**Fig. 5 | Validation of the importance of CCZ1 in MARV infection. a** Relative level of MARV infection in *CCZ1* KO, *MON1A* KO, *MON1B* KO. MARV infected cells were harvested for RNA extraction and subjected to qRT-PCR analysis. Data shown are viral RNA titers normalized to wild-type A549 cells. Error bar represent mean ± SD, $n = 2$ biologically independent replicates. **b** Reduction of infected cells number in *CCZ1* KO cells to wild-type cells determined by immunofluorescence. Error bar represent mean ± SD, $n = 4$ biologically independent replicates. **c** Relative level of infection of *CCZ1* KO to wild-type A549 cells infected with MARV, LASV or EBOV. Error bar represent mean ± SD, $n = 3$ biologically independent replicates. *$P < 0.05$, **$P < 0.01$, ***$P < 0.001$; two-tailed Student's *t*-test. Exact *p* values and Source data are provided as a Source Data file.

## CCZ1 controls filoviruses infections in human blood-vessel organoids

Blood vessels are one of the key organs for hemorrhagic filovirus infections in the human body. We therefore developed human blood-vessel organoids engineered from stem cells to assess filovirus infections in bona fide blood vessels that contain a lumen, endothelial cells, mural pericyte as well as a basal membrane[23]. To set up this infection model system, human blood vessels organoids were infected with EBOV for 3, 7 and 14 days to study the level of infection as well as the virus replication. As shown in Fig. 7a, EBOV is able to infect these 3D blood-vessel organoids and to replicate as determined by qRT-PCR. Next, we generated two different CRISPR/Cas9-mediated *CCZ1* knockout clones from the iPSC line NC8 (Fig. 7b) and generated human blood-vessel organoids. Wild-type and *CCZ1* mutated blood-vessel organoids were disaggregated and all the cells were further cultured as a monolayer in collagen-coated flasks. These cells were then infected with EBOV or MARV. In parallel, non-disaggregated blood-vessel organoids were also infected with EBOV or MARV. Importantly, *CCZ1* mutant blood vessels cells cultured as a monolayer (Fig. 7c) and, to a greater extent, bona fide blood-vessel organoids (Fig. 7d) showed a significant decrease in EBOV (monolayer: 85.25% for clone 1, 90.7% for clone 2; organoids: 99.5% for clone 1 and 2) and MARV (monolayer: 99.5% for clone 1, 99.6% for clone 2; organoids: 99.4% for clone 1 and

99.8% for clone 2) infection, visible by immunostaining (Supplementary Fig. 5), highlighting the critical role of CCZ1 in filoviruses infections in human stem cell derived blood vessels.

## CCZ1 knockout inhibits endolysosomal trafficking of MARV

Enveloped Filoviruses can enter cells through macropinocytosis and then undergo endolysosomal transport until the late endosome to reach the cytosol[24]. Our two top hits from VSVΔG/MARVGP screen that could affect MARV infection were RAB7 and CCZ1. The RAB7 GTPase is a late endosome marker and is essential for the fusion of late endosomes to lysosomes. RAB7 is recruited and then activated by the guanine nucleotide exchange factor CCZ1[25]. Its recruitment by the MON1-CCZ1 complex leads to the transition from early to late endosome[25–28]. To determine whether loss of *CCZ1* indeed affects the endolysosomal network, we performed a trafficking assay using DQ-Red BSA in A549 cells. In WT cells as well as *NPC1* KO cells, DQ-Red BSA traffics to lysosomes where it is cleaved by lysosomal hydrolases, resulting in a bright red fluorescent signal (Fig. 8a). By contrast, DQ Red BSA fluorescence signal was undetectable in *CCZ1* mutant cells (Fig. 8a), indicating that the loss of CCZ1, but not the loss of NPC1, interrupts endosomal trafficking ahead of lysosomes.

To determine whether the loss of *CCZ1* may affect entry mediated by filovirus glycoproteins, wild-type and *CCZ1* mutant A549 cells were

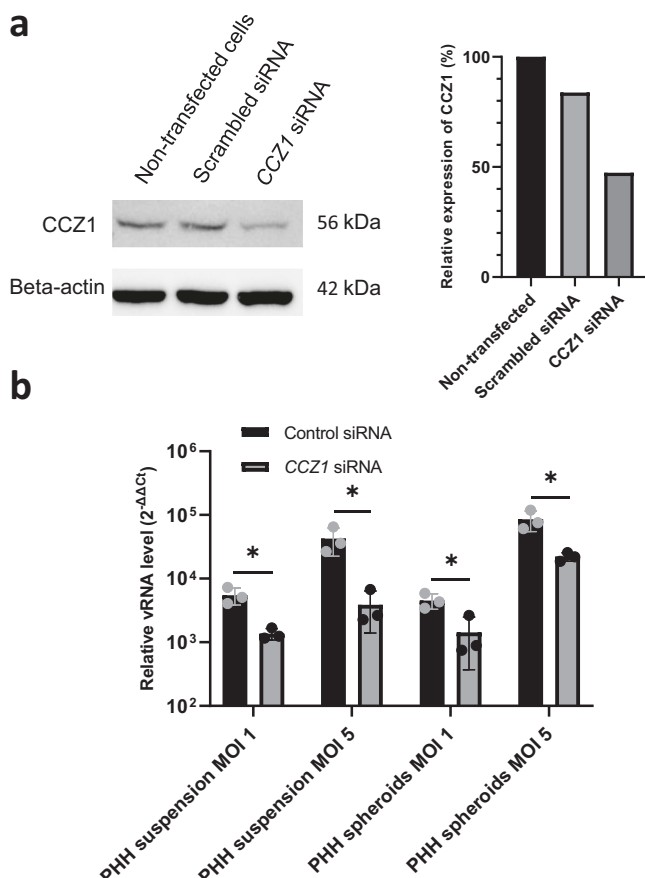

**Fig. 6 | Transient knockdown of *CCZ1* impairs VSVΔG/MARVGP infection in liver organoids. a** Western blot and associated densitometry analysis of the knockdown of *CCZ1* in Primary Human Hepatocytes(PHH). Uncropped blots in Source Data. **b** VSVΔG/MARVGP relative level of infection in PHH and 3D spheroids knocked down for *CCZ1* to control. Error bar represent mean ± SD, *n* = 3 biologically independent replicates. *$P < 0.05$; two-tailed Student's *t*-test. Exact *p* values and Source data are provided as a Source Data file.

infected with VSVΔG/MARVGP at a MOI of 20, fixed at an early time point post infection (6 h), and assessed for intracellular distribution of VSV-M protein. While WT cells exhibited diffuse cytoplasmic staining for VSV-M protein, indicative of normal infection and viral replication, we observed only punctate anti-M immunostaining in *CCZ1* mutant cells (Fig. 8b), revealing that the virus was trapped in the intracellular vesicular structures. Arenaviruses, like Lassa virus (LASV), are also known to use endolysosomal trafficking to reach the cytosol[29]. However, contrary to filoviruses, LASV enters the endolysosomal trafficking directly at the late endosome steps[29]. When *CCZ1* mutant cells were infected with VSVΔG/LASVGPC, the virus was able to infect the cells as indicated by a diffused cytoplasmic staining for the VSV-M protein (Fig. 8b). To determine if VSVΔG/MARVGP is stuck in early endosome when *CCZ1* is ablated, wild-type and *CCZ1* mutant A549 cells were infected with VSVΔG/MARVGP as previously described (MOI 20 for 6 h) and cells were stained for VSV-M protein as well as for RAB5, a marker of early endosome. The percentage of colocalization of these proteins was then measured in each condition. As shown in Fig. 8c, the percentage of virus in the early endosomes in *CCZ1* KO was significantly higher than in wild-type cells (Fig. 8c and Supplementary Fig. 6). As previously shown in Fig. 8b, wild-type cells exhibited diffused cytoplasmic staining for VSV-M protein contrary to *CCZ1* KO cells (Fig. 8c and supplementary fig. 5). These results indicate that CCZ1 controls filoviruses entry through early to late endosome trafficking.

## CCZ1 is important in endocytosis-dependent SARS-CoV-2 entry

To find factors important for SARS-CoV-2 infection, we runned a chemical mutagenesis haploid cells screening as previously described[30]. Briefly, AN3–12 haploid cells expressing human ACE2 at their surface were treated with N-ethyl-N-nitrosourea to induce random mutations in cellular genome. The obtained library were infected were SARS-CoV-2 and as above explained, cells resistant to infection survived. Each cell clone was grown separately and their full exome was sequenced to detect mutations common between several resistant clones. This screen highlighted ccz1 as a possible important factor for SARS-CoV-2 (Supplementary fig. 7). Moreover, it has been recently shown in a CRISPR screen, that CCZ1 plays a role in SARS-CoV-2 infection[31].

To confirm these data and better characterize the role of CCZ1 in SARS-CoV-2 infection, wild-type and *CCZ1* mutant A549 cells were infected with SARS-CoV-2. Loss of CCZ1 expression in A549 cells had no apparent effect on SARS-CoV-2 infection (Fig. 9a). A549 are characterized by low/null ACE2 expression levels, ACE2 being the critical entry receptor for all known SARS-CoV-2 variants[32–34]. In agreement with our previous data on filovirus and CCZ1-dependent endosomal pathway and knowing the ACE2 entry pathway TMPRSS2-independant involves endosomal trafficking[35], we knocked down (KD) *CCZ1* in Vero E6 cells, cells expressing a high level of ACE2[36,37] and not expressing TMPRSS2[38] (Fig. 9b). Infection of CCZ1 KD Vero E6 cells with SARS-CoV-2 lead to a 75% decrease in the level of SARS-CoV-2 infections (Fig. 9c). Finally, Vero E6 cells were knocked out for *CCZ1* (Supplementary Fig. 8), leading to a significant decrease (83.8%) of SARS-CoV-2 infection (Fig. 9d). These data demonstrate that CCZ1 is also involved in endocytosis-dependent SARS-CoV-2 infections via the endosomal pathway.

## Discussion

Emerging and re-emerging viral infections represent one of the largest threats to human and animal health, with a potential impact on the global economy. To date, only a few therapeutics against these pathogens have been approved to be used in humans. To develop antivirals and new therapeutic strategies to combat these infections, it is of paramount importance to better understand their replication cycle. In particular, the characterization of virus/host interactions will help us to better understand the biology and pathogenesis of viral infections and to identify targets for new therapeutic strategies.

The first phase of a viral infection of target cells is attachment and entry of viral particles. This step determines cell and tissue tropism and subsequent mechanisms of pathogenesis. Targeting the entry mechanisms is a hallmark of vaccines and certain antiviral therapeutics[24,39]. Classical approaches to characterize the entry factors for viruses rely on biochemical and immunological protocols, followed by the development of genetic methodologies based on complementary DNA libraries, microarrays and bioinformatics analyses, and more recently genome-wide screening using various RNA interference-based or CRISPR/Cas9 technologies[40]. In addition to these techniques, the development of haploid cell-based screening opened the opportunity to identify critical host factors for viral infections, as exemplified by NPC1 for filovirus infections or LAMP1 for LASV virus entry[7,8]. The discovery and characterization of NPC1 strongly advanced our knowledge about the mechanism by which filoviruses enter target cells showing that NPC1 interacts with glycoproteins to trigger the fusion between the viral envelope and the membrane of the late endosomes[41–43]. However, other factors must be involved in this process[42,44,45].

We utilized our previously established system of murine haploid stem cells to identify host factors that are critical for filovirus infections. Using this approach, we were able to identify various known and unknown factors involved in filovirus infections. Among the unknown host factors, CCZ1 was in the top hits. CCZ1, in complex with MON1 and other factors HOPS (homotypic fusion and vacuole protein sorting)

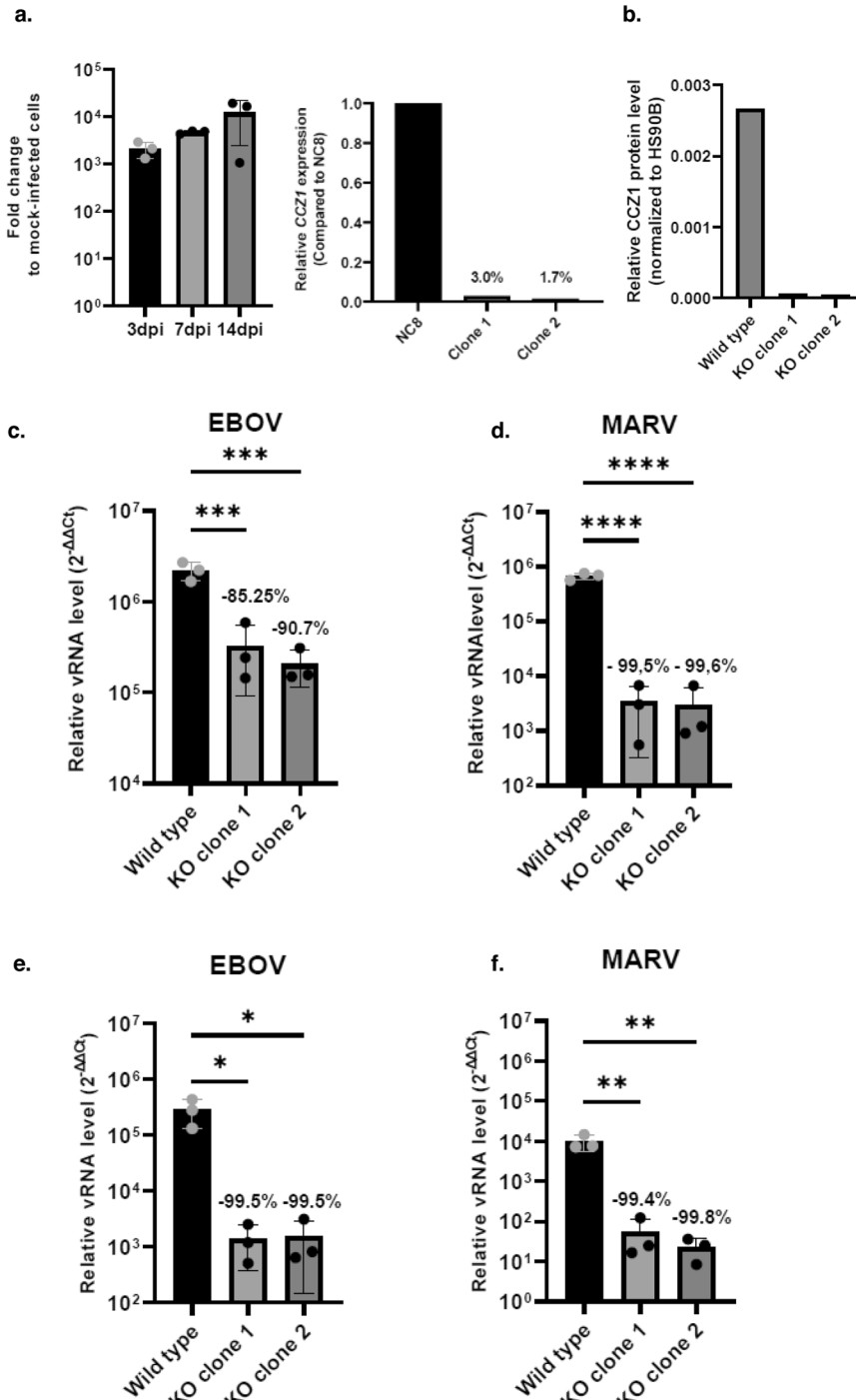

**Fig. 7 | *CCZ1* knockout reduces MARV and EBOV infection in blood vessels organoids. a** Infection and replication of EBOV in blood vessels organoids. Error bar represent mean ± SD, $n = 3$ biologically independent replicates. **b** Validation of the knockout of *CCZ1* in NC8 cells by measure of mRNA level and mass spectrometry **c** Relative level of infection of EBOV and MARV of 2D cultured blood vessels derived cells Error bar represent mean ± SD, $n = 3$ biologically independent replicates. ***$P < 0.001$, ****$P < 0.0001$. One-way ANOVA. **d** Relative level of infection of EBOV and MARV infection of blood vessels organoids. Error bar represent mean ± SD, $n = 3$ biologically independent replicates. *$P < 0.05$, **$P < 0.01$. One-way ANOVA. Exact $p$ values and Source data are provided as a Source Data file.

tethering complex, mediates RAB5 to RAB7 conversion, mediating the maturation of early to late endosomes[28]. It also acts with MON1 as a guanine exchange factor for RAB7, activating RAB7 for the fusion between late endosomes and lysosomes[28]. Filoviruses (EBOV and MARV) are known to enter the cells using the early to late endosome pathway[4,24,46] and the HOPS complex was shown to be an essential factor in EBOV entry[6]. Indeed, using multiple systems from cells to blood-vessel organoids, we were able to confirm that CCZ1

downregulation inhibits VSVΔG/MARVGP infections, trapping the virus inside early endosomal vesicles. Importantly, experiments conducted with authentic filoviruses showed that CCZ1 is indeed required for MARV and EBOV infections, supporting a common pattern for intracellular trafficking of filoviruses and making CCZ1 an interesting target for the development of antivirals. Importantly, inactivation of CCZ1 had no effect on infection with LASV. LASV is also responsible of severe hemorrhagic fevers and enters into the targets cells by an

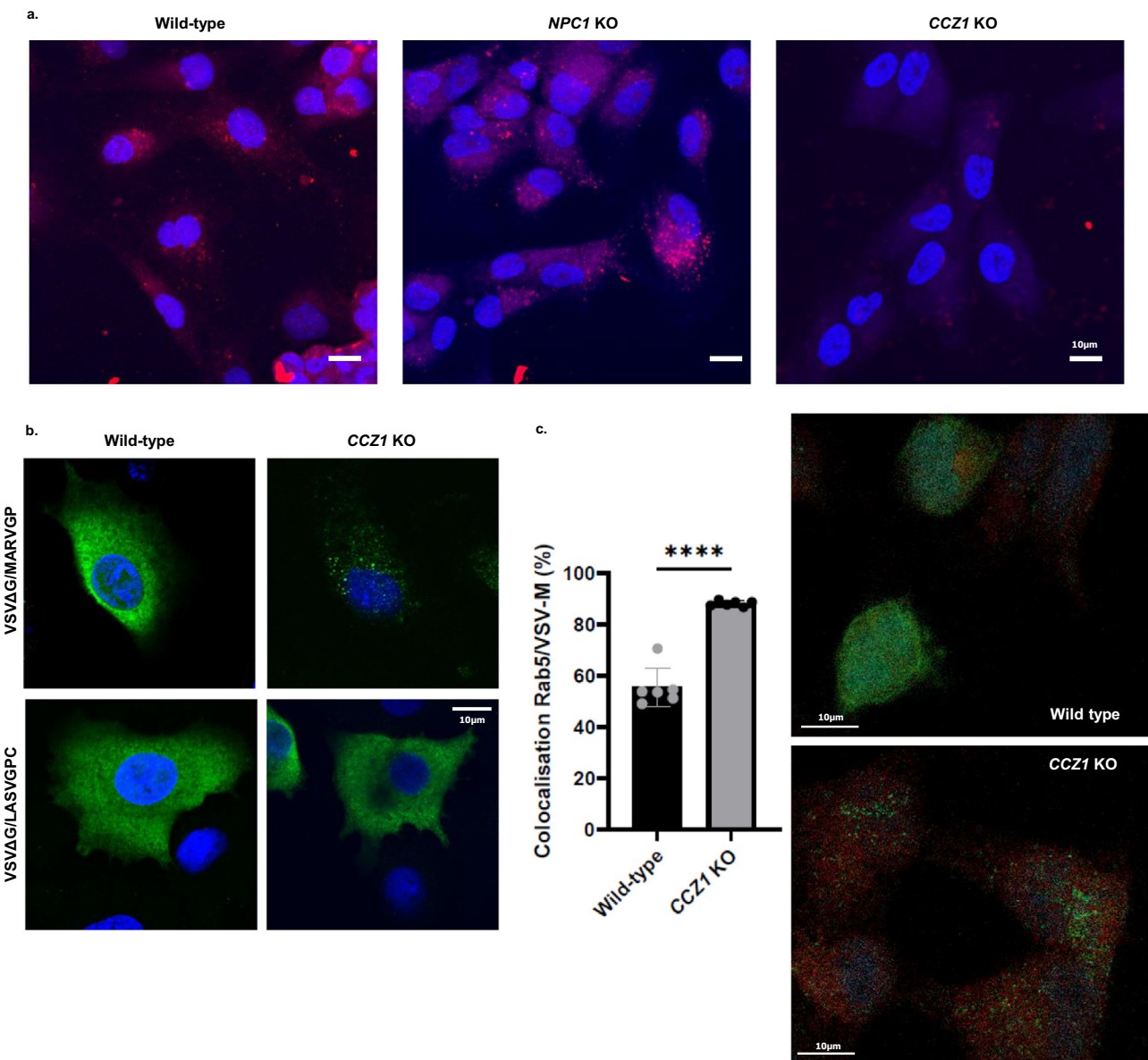

**Fig. 8 | Effect of the loss of CCZ1 on virus trafficking in the endosomal pathway.**
**a** Microscopy of CCZ1 of DQ-Red BSA treated cells. In *CCZ1* KO cells, DQ-Red BSA doesn't reach lysosomes (Scale 10 μm). Pictures are representative of 3 independent experiments. **b** *CCZ1* KO leads to a blocking of VSVΔG/MARVGP in vesicles (Green: VSV-M protein, Blue: DAPI). Pictures are representative of 3 independent experiments. **c** Colocalization of VSV-M protein with the early endosome marker

Rab5 in VSVΔG/MARVGP infected wild-type and *CCZ1* KO A549. Green: VSV-M protein, Red: Rab5, Blue: DAPI. (Left: level of colocalization; Right: confocal microscopy images). All single channels are presented in supplementary Fig. 5. Error bar represent mean ± SD, *n* = 6 cells. ****P < 0.001; two-tailed Student's *t*-test. Exact *p* values and Source data are provided as a Source Data file.

unknown mode of endocytosis, reaching directly the late endosomes via multivesicular bodies[29,47,48] without going through early endosome, explaining the lack of effect of the knockdown and knockout of *CCZ1* on LASV infections. Nevertheless, it is important to note that the levels of inhibition achieved with PHH and siRNA were not as high compared to the other cell lines. This could be attributed to the potentially lower transfection efficiency of siRNA in human primary cells. However, it is also possible that the virus utilizes distinct pathways for entry in these cells, leading to differences in the observed inhibition levels. Capthesins are required for EBOV entry, as they cleave the viral glycoproteins to induce the fusion of the viral membrane with the late endosome membrane and the release of the viral genome into the cytoplasm[4]. The spike protein of SARS-CoV-2 can also be processed by endosomal capthesin B and L in addition to TMPRSS2[49]. It was recently shown that the major histocompatibility complex class II transactivator (CIITA), a

transcription factor, is able to disrupt the capthesin-mediated EBOV glycoproteins and SARS-CoV-2 spike proteins[50]. SARS-CoV-2 uses different pathways to enter cells, mainly of them being ACE2-dependent. In ACE2-dependent entry, two mains pathways are used by the virus to enter the cells. When TMPRSS2 is present and active, it activates the viral spike protein and the viral envelope fuses with the plasma membrane, releasing the viral genome into host cells without the need of entering the endosomal pathway[38,51]. When TMPRSS2 is absent or inactive, the virus enters the cells via the endosomal pathway[38,51]. A549 cells do not express ACE2 neither TMPRSS2 and the knockout of *CCZ1* doesn't affect SARS-CoV-2 entry, suggesting the entry pathway used by SARS-CoV-2 in A549 (that is independent of ACE2 and TMPRSS2) does not involve the early to late endosome endocytosis. Infections of Vero E6 cells, expressing ACE2 but not TMPRSS2, lead to an endosomal trafficking of the virus. In these cells, the knockdown and knockout of

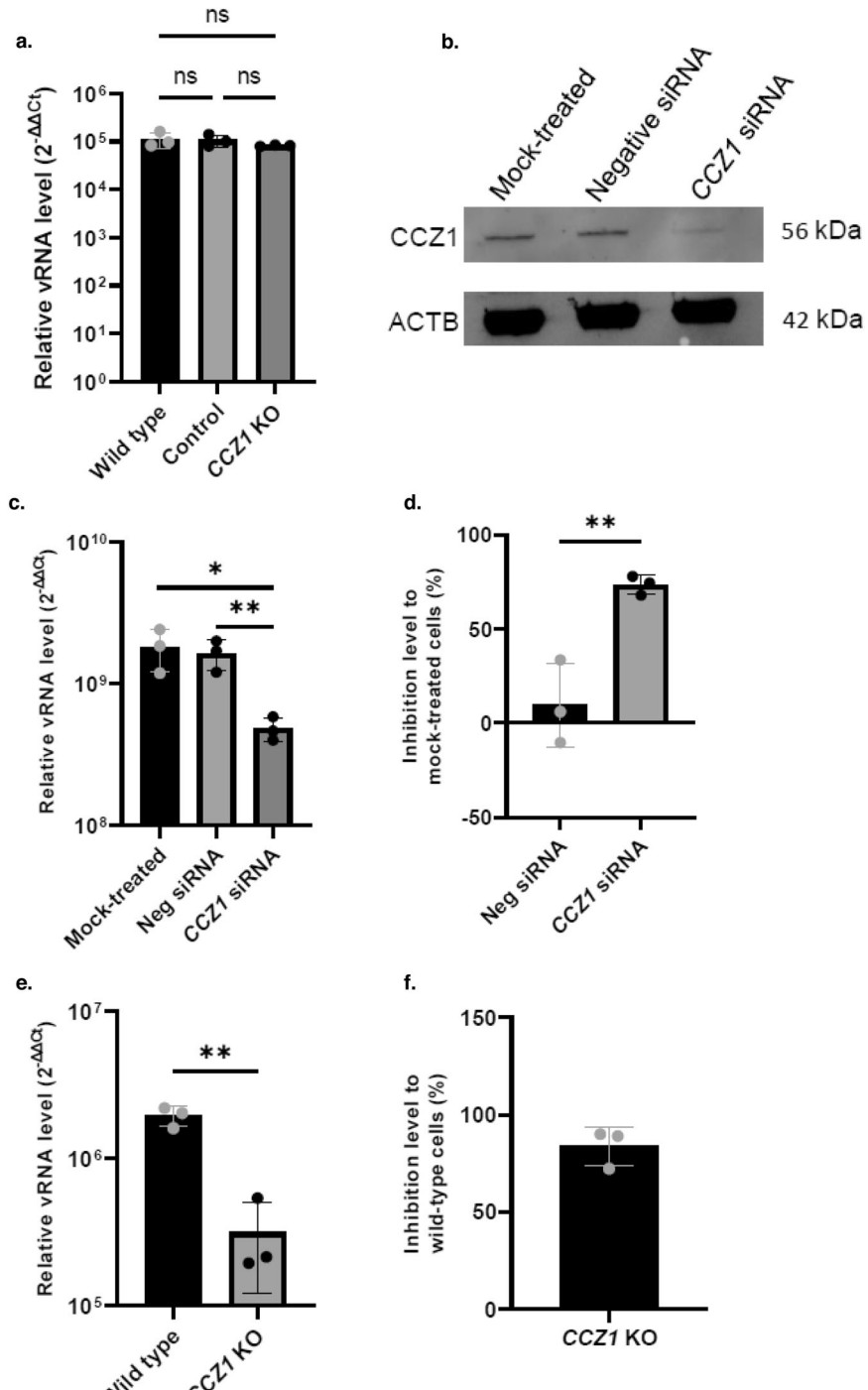

**Fig. 9 | Role of CCZ1 in SARS-CoV-2 infection. a** Relative level of infection of *CCZ1* KO A549 cells by SARS-CoV-2. Error bar represent mean ± SD. *n* = 3 independent experiments. Two-tailed Student's *t*-test **b** Western blot analysis of CCZ1 protein expression in siRNA transfected Vero E6. Uncropped blots in Source Data. **c** Relative level of infection and **d** inhibition of *CCZ1* knocked down Vero E6 cells by SARS-CoV-2. Error bar represent mean ± SD, *n* = 3 biologically independent replicates. *P < 0.05, **P < 0.01; Two-tailed Student's *t*-test. **e** Relative level of infection and **f** inhibition of *CCZ1* KO Vero E6 cells by SARS-CoV-2. Error bar represent mean ± SD, *n* = 3 biologically independent replicates. **P < 0.01; Two-tailed Student's t-test. Exact *p* values. Source data are provided as a Source Data file.

*CCZ1* affect SARS-CoV-2 entry. These data highlight CCZ1 as a target for antivirals in infectious diseases related to viruses using the endosomal pathway for infection.

In conclusion, we developed a platform for the identification of host factors necessary for filovirus infection and for their validation using mutant cells, hepatocyte spheroids and blood-vessel organoids. Using this approach, we identified CCZ1 as a key factor for the specific entry of filoviruses such as Marburg and Ebola viruses.

Moreover, *CCZ1* downregulation affects endocytosis-dependent SARS-CoV-2 infections, suggesting that CCZ1 controls a common endosomal pathway for several virus infections. Targeting CCZ1 could potentially serve as a promising drug target for controlling infections caused by various viruses, such as SARS-CoV-2, Marburg, and Ebola. However, it is important to emphasize that currently there are no available drugs specifically designed for this purpose.

## Methods

### Cells and viruses

Haploid mSCs AN3–12 are a feeder independent clonal derivative of HMSc2 isolated from mice oocyte and maintained at IMBA[52]. The AN3–12 library and knocked out cells used for the haploid screening was obtained from IMBA (Austria)[14,52]. AN3–12 cells were validated by STR analysis. Haploid mES cells were maintained in standard EScell medium, supplemented with 10% (v/v) fetal bovine serum (Hyclone), recombinant mouse Leukemia Inhibitory Factor (LIF) and β-mercaptoethanol. A549 cells subjected to CRISPR/Cas9 (No-target control, GFP control, CCZ1, NPC1 and RAB7) were obtained from IMBA (Austria). Original A549 cells were bought from ATCC (CCL-185), validated by ATCC (https://www.atcc.org/products/ccl-185) and used at early passage (<10). Cells were cultured in Dulbecco's modified Eagle's medium (D-MEM, Gibco) with 10% (v/v) fetal bovine serum (Hyclone). VSVΔG/MARVGP Musoke strain and VSVΔG/LASVGPC were produced as previously described[15]. Marburg virus (MARV) Musoke strain (GenBank accession number DQ217792), Ebola virus (EBOV) Zaire strain and Lassa virus (LASV) Josiah strain were cultured on Vero cells. NC8 cells were kindly gifted by Prof. Manfred Boehm (NIH) and validated by Multiplex-fluorescence in situ hybridization and whole-genome SNP genotyping[23]. SARS-CoV-2 Wuhan-Hu-1 strain (Pango lineage B; GenBank: MT093571) was cultured on Vero E6 cells (ATCC C1008). Briefly, Vero E6 were infected with SARS-CoV-2 at a MOI of 0.1 for 3 days. After 3 days, supernatant was recovered, centrifuged and aliquoted before being titered[33].

### Biosafety

All our experiments involving EBOV, MARV and LASV were done in biosafety level 4 laboratory in compliance with the Swedish Public Health Agency guidelines (Folkhälsomyndigheten, Stockholm).

### Plasmids

Plasmids pCAGGS_CCZ1, pCAGGS_MON1A, and pCAGGS_MON1B were made by gene synthesis (BioCat GmbH), based on the published sequences NM_015622.6 (human CCZ1: https://www.ncbi.nlm.nih.gov/nuccore/NM_015622.6), NM_032355.4 (human MON1A: https://www.ncbi.nlm.nih.gov/nuccore/NM_032355.4), and NM_014940.4 (human MON1B: https://www.ncbi.nlm.nih.gov/nuccore/NM_014940.4), respectively. A Kozak sequence was added in 5′, and the genes were cloned into the pCAGGS vector (gifted by Prof Friedemann Weber, Giessen, Germany), digested by XhoI and NotI (NEB). Competent bacteria were transformed (DH10B, Thermo Fisher), plasmids were prepared using the Qiagen Plasmid midi-kit, and the sequence of the genes were validated (Eurofins).

### Screen for MARV and LASV host factors using Haploid mouse stem cells

Isolation of VSVΔG/MARVGP resistant haploid mSCs was performed as described previously[52,53]. In brief, 100 million haploid mSCs were infected with VSVΔG/MARVGP or VSVΔG/LASVGPC at a MOI of 1 in 0.5 ml of ES medium without FBS. One hour after infection, the cells were supplemented with complete ES medium and incubated at 37 °C with 5% $CO_2$. After growth, the virus-resistant cells were pooled and the genomic DNA of these resistant colonies was isolated and subjected for the sequencing protocol at the sequencing facility in IMBA (Austria).

### Analysis of gene-trap insertion sites by sequencing

Insertion sites in MARV and LASV resistant populations of mSCs were determined by sequencing the flanking regions of the gene traps insertion sites as previously described[14]. Briefly, DNA was isolated surviving cells and processed for linear PCR followed by linker ligation, PCR and next generation sequencing methods as described previously[14,52,54]. Significant genes were selected using a binomial test

of disruptive insertions compared to the undisruptive insertions and was done for each gene as described previously[14,52,54,55]. Insertion sites in significant gene hits were identified and the p values for enrichment were determined using the Fishers exact test[52]. Raw data are presented in Source data.

### Infection of haploid cells for validation of the ccz1, npc1 and rab7

AN3–12 cells were knocked out for ccz1, npc1 and rab7 by insertional mutagenesis. Inverted sister clones were produced by the insertion of the gene-trap as previously described[14,52]. All cells were produced by Haplobank (IMBA). Wild type, ccz1, npc1 and rab7 KO cells as well as their inverted sister clones were infected with VSVΔG/MARVGP at a MOI of 0.1 for 1 h at 37 °C in EScell medium 0% FBS. After one hour, cells were washed once with PBS and EScell medium 15% FBS was added to each well. Twenty-four hours post infection, cells were washed 3 times with PBS and lyzed using trizol. Viral RNA level was assessed by qRT-PCR and the result was normalized to an endogenous gene (RNase P). Relative vRNA level was measured by $2^{-\Delta\Delta Ct}$ method.

### Infection of A549 with VSVΔG/MARVGP

A549 WT, control, CCZ1$^{KO}$, NPC1$^{KO}$ and RAB7$^{KO}$ cells were infected with VSVΔG/MARVGP at a MOI of 5 for 48 hpi. Cell extracts were prepared for Western blotting as described below.

### Flow cytometry

A549 WT, CCZ1$^{KO}$ (clone 2), NPC1$^{KO}$ (clone 2) and RAB7$^{KO}$ (clone 3) were mixed with GFP expressing wild-type cells at a ratio of approximately 30% KO to 70% wild-type cells in DMEM (Thermofisher) containing 10% FBS (Gibco). A total of $5 \times 10^5$ cells per well were plated in 6 well plates and incubated at 37 °C with 5% $CO_2$ overnight. Medium was removed and the cell mixtures was either mock infected or infected with VSVΔG/MARVGP at a MOI of 5 in a total volume of 0.5 ml. After 1 h incubation at 37 °C, 1.5 ml of complete medium was added to each well. Infected and mock-treated cells were harvested at different time points (48, 72 and 120 h post infection) using trypsin and resuspended in DMEM medium. After centrifugation at $200 \times g$ for 5 min, the supernatant was removed and cells were fixed in 4% paraformaldehyde for 30 min at 4 °C to inactivate the virus. After the fixation step, cells were pelleted by centrifugation and resuspended in 400 µl of flow buffer. The percentage of GFP positive and negative cells in the cell mixtures at each time point was measured using a LSRII flow cytometry. All flow cytometry data were analyzed using the Flow Jo software.

### CCZ1, MON1A and MON1B assay

CCZ1, MON1A and MON1B KO A549 were transfected with plasmids coding for CCZ1, MON1A and MON1B, respectively. Briefly, for one well, 500 µg of each plasmid was mixed with 1 µl of GeneJammer (Agilent) in 50 µl of Opti-MEM and incubated at room temperature for 20 min. Fifty microliters of the mix was deposited in a well of 48-well plates and $5 \times 10^4$ A549 KO cells in 250 µl of DMEM 5% FBS were added on top of the plasmid mix. Cells were incubated during 3 days at 37 °C. Three days post transfection, cells were counted and infected with VSVΔG/MARVGP at a MOI of 0.1 in DMEM 2% FBS for 1 h. Cells were then washed once with PBS and DMEM 5% FBS was added. Cells were incubated at 37 °C for 48 h. Cells were then washed 3 times with PBS and the cells harvested to be analyzed by qRT-PCR. and normalized to the RNA level of endogenous genes. Relative vRNA level was measured by $2^{-\Delta\Delta Ct}$ method.

### Infection of A549 with Marburg virus

CRISPR/Cas9-mediated knockout of CCZ1, NPC1 and RAB7 in A549 cells as well as wild-type cells were seeded at $5 \times 10^4$ cells per well in a 24 well plate (Corning). The following day, cells were counted and were infected with Marburg virus at a MOI of 0.1 for 1 h. One hour post infection, the inoculum were removed, cells washed once with PBS and

500 μl of DMEM 10% FBS were added per well. Forty-eight hpi, the supernatants were removed, the cells were washed 3 times with PBS, then lysed using 300 μl of Trizol™ (ThermoFisher). RNA was extracted and analyzed by qRT-PCR as described below and normalized to the RNA level of endogenous genes ($2^{-\Delta\Delta Ct}$). The resulting values were then expressed in percentage of infection to wild-type infected cells.

### Infections of A549 with EBOV, LASV and MARV

A549 WT, control and *CCZ1* KO (clone 2) cells were seeded in 24 well plate at the density of $5 \times 10^4$ cells per well for 24 h. After 24 h, cells were infected with MARV, EBOV, LASV virus at a MOI of 0.1 for 1 h. One hour post infection, the inoculum were removed, cells washed once with PBS and 500 μl of DMEM 10% FBS were added per well. Total RNA was isolated from the infected cells after 48 h and the level of viral RNA in each sample was measured by qRT-PCR and normalized to the RNA level of endogenous genes. Relative vRNA level was measured by $2^{-\Delta\Delta Ct}$ method.

### Transient transfection of hepatocytes with CCZ1 siRNAs

Primary Human Hepatocytes (PHH), obtained from Bio IVT (USA) were seeded in ultra-low attachment plates and transfected with 50 nM of CCZ1 siRNAs (On-TARGETplus SMARTpool Human CCZ1 Cat# L-021482-02-0005) or control siRNA (AllStars Neg. Control siRNA, QIAGEN) per well as previously described[21]. For western blot analysis, cell lysates were prepared and the endogenous levels of CCZ1 were analyzed 72 h post transfection as described below. For infection, PHH were infected in suspension 3 days post seeding or in 3D spheroids 5 days post seeding.

### Infection of primary human hepatocytes in suspension or in 3D liver spheroid with VSVΔG/MARVGP

PHH in suspension and in 3D spheroids were prepared as described above and infected with VSVΔG/MARVGP at a MOI of 1 or 5 in William's medium for 24 h. Twenty-four hours post infection, 1/3 of plates of cells/organoids were pooled per condition, washed 3 times with PBS, lysed with Trizol and total RNA was extracted and analyzed by qRT-PCR and normalized to the RNA level of endogenous genes. Relative vRNA level was measured by $2^{-\Delta\Delta Ct}$ method.

### Infection of blood-vessel organoids for model validation

Blood-vessel organoids were produced as previously described[56] and were infected with $10^3$ FFU of Ebola Zaire strain per well in blood vessels sprouting media. The plate was covered with a breathable seal (Nunc, Thermofisher) to limit evaporation and incubated on an orbital shaker for 3, 7 and 14 days. Three organoids per condition were then pooled, washed 3 times with PBS and lysed with Trizol. RNA was extracted and analyzed by qRT-PCR and normalized to the RNA level of endogenous genes. Relative vRNA level was measured by $2^{-\Delta\Delta Ct}$ method.

### Generation of *CCZ1/CCZ1B* KO NC8 iPSCs

gRNAs targeting both paralogs of *CCZ1* (*CCZ1* and *CCZ1B*) were designed using CHOPCHOP (http://chopchop.cbu.uib.no/). Selected gRNA was cloned into the all-in-one PX459 v2.0 plasmid as previously described[57]. The plasmid was electroporated into the stem cells by mixing $10^6$ cells with 500 ng of plasmid, followed by electroporation using an Amaxa 4D nucleofector. Following electroporation, cells were replated and expanded in 1 μg of puromycin before colony picking. Following colony picking, clones were validated for knockout of CCZ1 and CCZ1B at the DNA level using Sanger sequencing and TIDE analysis (http://shinyapps.datacurators.nl/tide/) and qRT-PCR.

### Validation of *CCZ1* knockouts using mass spectrometry.

Peptide libraries for mass spectrometry (MS) analysis were prepared from cell pellets of *CCZ1*+/+ and *CCZ1*−/− stem cells using the PreOmics iST kit according to the manufacturer's instructions.

### Relative peptide amount determination.

Final peptide amounts were determined by separating an aliquot of each sample on a liquid chromatography-UV system equipped with a monolith column based. The final amounts were determined through comparison to a peak area of 100 ng of Pierce HeLa protein digest standard (PN 88329; ThermoFisher Scientific). Peptide solution was frozen at −70 °C before further processing.

### NanoLC-MS/MS analysis and data processing.

The nano high pressure liquid chromatography system used was an UltiMate 3000 RSLC nano system coupled to the Orbitrap Exploris 480 mass spectrometer, equipped with a NanoFlex nanospray source (Thermo Fisher Scientific). Peptides were loaded onto a trap column (Thermo Fisher Scientific, PepMap C18, 5 mm × 300 μm ID, 5 μm particles, 100 Å pore size) at a flow rate of 25 μL/min using 0.1% TFA as the mobile phase. After 10 min, the trap column was switched to an analytical column (Thermo Fisher Scientific, PepMap C18, 500 mm × 75 μm ID, 2 μm, 100 Å). The analytical column was connected to PepSep sprayer 1 (Bruker) equipped with a 10 μm ID fused silica electrospray emitter with an integrated liquid junction (Bruker, PN 1893527). Electrospray voltage was set to 2.4 kV. Peptides were eluted using a flow rate of 230 nl/min, and a binary 120 min gradient. The gradient starts with the mobile phases: 98% A (water/formic acid, 99.9/0.1, v/v) and 2% B (water/acetonitrile/formic acid, 19.92/80/0.08, v/v/v), increases to 35% B over the next 120 min, followed by a 5 min gradient resulting in 90% B. The gradient is maintained for 5 min, and subsequently reverts over 2 min to the original gradient 98% A and 2% B for equilibration at 30 °C. The Orbitrap Q-ExactiveHF-X mass spectrometer was operated by a mixed MS method which consisted of one full scan (*m/z* range 380−1500; 15,000 resolution; AGC target value 3e6) followed by the PRM of targeted peptides from an inclusion list (isolation window 0.8 *m/z*; normalized collision energy (NCE) 30; 30,000 resolution, AGC target value 2e5). The maximum injection time was set to 800 ms.

A scheduled PRM method (sPRM) development, data processing and manual evaluation of results were performed in Skyline[58] (64-bit, v22.2.). For the qualitative identification of the CCZ1 protein were used 9 unique trypsin specific peptides. Three other proteins from the human proteome were selected for quantification of CCZ1 amounts in the original samples. Details about all peptides included in the sPRM method are stated in Source Data.

### Blood-vessel organoids dissociation and replating

Twenty to 25 mature organoids per condition were transferred into prefiltered and prewarmed enzymatic dissociation mix consisting of 4 mg Liberase TH (Sigma Aldrich), 30 mg Dispase II (Life Technologies) and 350 μl DNAseI dissolved in 10 ml of PBS. The organoid containing enzymatic mix was incubated for 20 min at 37°C followed by trituration 15 times with a p1000 pipette. The 37 °C incubation and trituration were repeated twice more. The dissociated organoids were passed through a 70 μm cell strainer into 5 ml of cold DMEM/F12 medium. Following filtering, the cells were collected through centrifugation (300 × *g*, 5 min) and replated in T-25 flasks at 44,000 cells/cm² in sprouting media (StemPro-34 SFM medium, 1X StemPro-34 nutrient supplement, 0.5 ml Glutamax, 15% FCS, 100 ng/ml VEGF-A and 100 ng/ml FGF-2).

### Infection of blood-vessel organoids and dissociated organoids for CCZ1 KO assay

Blood-vessel organoids of wild-type and *CCZ1* KO genotype were infected with $10^3$ FFU of Ebola virus or Marburg virus per well in blood vessels sprouting media[56]. The plates were covered with a breathable seal (Nunc, Thermofisher) and incubated on an orbital shaker for

3 days. Three organoids per condition were then pooled, washed 3 times with PBS and lysed with Trizol (Thermofisher). RNA was extracted and analyzed by qRT-PCR and normalized to the RNA level of endogenous genes. Relative vRNA level was measured by $2^{-\Delta\Delta Ct}$ method.

For dissociated organoids, $5 \times 10^4$ cells per well were seeded in a 48-well plate. Twenty-four hours post seeding, cells were counted and infected at a MOI of 0.1 with Ebola virus or Marburg virus in sprouting media for 1 h. One hour post infection, cells were washed once with PBS and 500 μl of sprouting medium per well were added. Cells were incubated for 3 days at 37 °C 5% $CO_2$. Three days post infection, cells were washed 3 times with PBS, lysed with Trizol and analyzed by qRT-PCR and normalized to the RNA level of endogenous genes. Relative vRNA level was measured by $2^{-\Delta\Delta Ct}$ method.

### Immunofluorescence assay of blood-vessel organoids
Blood-vessel organoids (BVO) were cryo-embedded in OCT molds submerged in isopropanol dry ice bath until frozen. Then, the organoids were sectioned at 8 μm thickness using CryoStar NX70 cryostat (Epredia). For immunofluorescence staining, the sections were washed twice with PBS for 10 min at room temperature (RT) before adding PBTA buffer (5% BSA, 0.25% Triton X-100, 0.01% NaN3 in PBS), as a blocking step, for two hours at RT. Subsequently, the sections were incubated overnight at 4 °C with primary antibodies targeting Ebola virus Glycoproteins (home-made, Rabbit, 1/100), VE-Cadherin (Santa-Cruz, Clone F8, #sc-9989, Mouse, 1/50) and PDGFR (Bio-Techne #AF385, Goat, 1/100) in PBTA buffer. On the second day, the samples were washed 3 times 15 min with PBS at RT, then incubated with the corresponding secondary antibody diluted in PBTA 1:500 for 2 h at RT (Goat anti-rabbit 488 (Thermofisher, #A32731), donkey anti-mouse 555 (Thermofisher, #A32773), rabbit anti-goat 647 (Thermofisher, #A21446)). As a final step, the sections were washed 3 × 15 min with PBS at RT and mounted using Prolong Gold Antifade mounting reagent with DAPI (ThermoFisher). Images were obtained using a Zeiss LSM880 confocal microscope.

### Immunofluorescence assay for in situ VSVΔG/MARVGP localization
A549 WT and CCZ1 KO cells, were infected with VSVΔG/MARVGP at an MOI of 20. After 6 h, cells were fixed using 4% paraformaldehyde at 4 °C and incubated with primary anti-VSV-M antibody (Kerafast, clone 23H12, #EB0011) at 1:2000 dilution in immunofluorescence buffer (BSA 0.2%, Triton X-100 0.1% in PBS, pH 7.4) for 30 min at 37 °C. Cells were washed with PBS before incubation with secondary antibody (Goat anti-mouse IgG Alexa Fluor™ Plus 488 antibody, ThermoFisher, #A32723) (1:1000) and DAPI (1:1000). Images of the infected cells was captured using confocal laser scanning microscope (Zeiss LSM 800).

### Lysosomal activity assay using DQ-red BSA
To check the lysosomal activity, A549 WT, CCZ1 KO, and NPC1 KO, cells were grown on chamber slides for 24 h. After 18–24 h, the cells were treated with DQ-red-BSA (Thermofisher) 10 μg/ml and incubated at 37 °C for 4 h. Subsequently, cells were washed with 1× PBS to remove excess dye, fixed with 2% paraformaldehyde and the images were acquired using confocal laser scanning microscope with an oil immersion objective (Zeiss LSM 800).

### Colocalization of Rab5 and VSVΔG/MARVGP M protein
A549 wild-type and CCZ1 KO cells were seeded in Chamber Slide System (Nunc Lab-Tek, Thermofisher). Twenty-four hours post seeding, cells were infected with VSVΔG/MARVGP at a MOI of 20 in DMEM 2% FBS. One hour post infection, the inoculum was removed, cells were washed once with PBS and DMEM 10% FBS was added. Six hpi, supernatant was removed. The cells were washed 3 times with PBS and fixed using 4% paraformaldehyde at 4 °C. The cells were then incubated with primary anti-VSV-M antibody (Kerafast, clone 23H12, #EB0011) at 1:2000 dilution and with anti-Rab5 antibody (Thermofisher, #PA5-29022) at 1:1000 dilution in immunofluorescence buffer (BSA 0.2%, Triton X-100 0.1% in PBS, pH 7.4) for 30 min at 37 °C. Cells were washed with PBS before incubation with secondary antibodies (Goat anti-mouse IgG Alexa Fluor™ Plus 488 antibody, 1:1000, ThermoFisher, #A32723 and Goat anti-Rabbit IgG Alexa Fluor Plus 594 antibody, 1:1000, Thermofisher, #A32740) and DAPI (1/1000). Images of the infected cells were captured using a confocal laser scanning microscope (Zeiss LSM 800). Images were analyzed for colocalization between green (VSV-M) and red (Rab5) signals using ImageJ software[59]. Results were presented using GraphPad Prism version 9.4.1 for Windows (GraphPad Software, San Diego, CA, USA, www.graphpad.com).

### Chemical mutagenesis of haploid stem cells
Chemical mutagenesis using N-Ethylnitrosurea (ENU) was performed as described previously[30]. Briefly, haploid AN3–12 cells overexpressing human ACE2, were treated, in suspension and under constant agitation, with 0.1 mg/ml ENU in full medium while for 2 h. Cells were then washed 5 times and transferred to a culture dish. Cells were left to recover for 48 h. Cells were then detached and singled using Trypsin/EDTA before being frozen in 10% DMSO, 40% FBS and 50% full medium.

**Chemically mutagenized haploid cells screening.** 50 million of mutagenized haploid mSCs were infected with SARS-CoV-2 at a MOI of 5 in 5 ml of ES medium without FBS in 150 mm dish. One hour after infection, the cells were supplemented with complete ES medium and incubated at 37 °C with 5% $CO_2$. After outgrowth of virus-resistant cells, cell clones were picked separately and cultured. Resistance to SARS-CoV-2 was validated by infection with SARS-CoV-2. $5 \times 10^4$ wild-type AN3–12 and potentially resistant clones cells per well (48-well plates) were seeded in complete ESCell medium. Four hours post seeding, cells were infected with SARS-CoV-2 at a MOI of 5 and checked for cell survival for a week. All clones resistant to SARS-CoV-2 infection were subjected to DNA extraction using the Gentra Puregene Tissue Kit (Qiagen). Paired end, 150 bp whole exome sequencing was performed on an Illumina Novaseq 6000 instrument after precapture-barcoding and exome capture with the Agilent SureSelect Mouse All Exon kit. For data analysis, raw reads were aligned to the reference genome mm9. Variants were identified and annotated using GATK and snpEff. CCHFV resistance causing alterations were identified by allelism only considering variants with moderate or high effect on protein and a read coverage >20.

### CCZ1 knockdown and infection of Vero E6 cells
Vero E6 cells were reverse-transfected with CCZ1 siRNA (Thermofisher, cat#s230589 and cat#s230591) using Lipofectamine RNAimax, following company's instructions for 24 well plates. Seventy-two hours post transfection, cells were either lysed for subsequent analyses by western blot (see below) or were infected with SARS-CoV-2 (Genbank accession MT093571) at a MOI of 0.1 in DMEM 5% FBS for 6 h. Six hours post infection, cells were washed with PBS 3 times before being lysed with Trizol for subsequent analyses by qRT-PCR.

### CCZ1 knockout generation in Vero E6 cells
CCZ1 gene was edited using CRISPR-Cas9 technology. sgRNA GCTGTGTGCTTTATGATCGA was chosen to target CCZ1 locus. The sgRNA sequence was cloned into a lentiviral transfer plasmid containing Cas9 and a puromycin resistance cassette.

Lentiviral particles were produced and Vero E6 cells were transduced. After selection with puromycin, Western Blot was used to probe for CCZ1 protein expression levels using an anti-CCZ1 antibody (1/500, #HPA050006, Sigma-Aldrich). As a loading control Anti-Beta-

actin antibody was used (1/1000, clone 8H10D10, #3700, Cell Signaling Technology).

## qRT-PCR

All RNA extractions were performed using Direct-zol RNA extraction kit (Zymo Research)following the manufacturer protocol. Quantitative real time PCR reactions were performed using a TaqMan Fast Virus 1-step Master Mix (Thermofisher) and run on an Applied Biosystems machine. Following primers were used in this study to detect MARV VP40 Fwd: GCGTATAACGARCGAACAGTCA Rev: AGCCACAGTATG RGCTARTATTC probe: FAM-AAATTGCTCATRATCCCRAGAGGCAGCC A-TAMRA, EBOV GP gene[60] or LASV S gene (Fwd: CCATTCCTAACTT-CAATCARTATGARGCAATGAG; Rev: GGTCTAGAAAACTGGCAGTGAT CTTCCCA; Probe: FAM-ATGAGRATGGCNTGGG-MGB), VSV-M gene (Fwd: TGATACAGTACAATTATTTTGGGAC; Rev: GAGACTTTCTGTTA CGGGATCTGG; Probe: FAM-ATGATGCATGATCCAGC-MGB) or SARS-CoV-2 E gene as previously described[33]. RNase P RNA was used as an endogenous control for normalization. Relative vRNA level was measured by $2^{-\Delta\Delta Ct}$ method.

## Western blot

Cell pellets were lysed using a lysis buffer containing 10 mM of Tris, 150 mM of NaCl, 0.5% of SDS and 1% Triton X-100 in PBS, pH 7.5 (Karolinska Institutet) and mixed with LDS Sample Buffer (NuPAGE, Invitrogen) containing 10% beta-mercaptoethanol (Sigma-Aldrich). Lysates were boiled at 98 °C for 20 min. Protein concentration in each sample was measured using Pierce™ 660 nm Protein Assay Reagent (Thermofisher) associated with Ionic Detergent Compatibility Reagent (Thermofisher). Twenty micrograms of protein were deposited in each well of a 4–12% Criterion Bis-Tris gel (Bio-Rad). PVDF Membranes were blocked using 5% milk in 0.1% Tween (PBS-T) for one hour. The membranes were incubated with primary antibodies (see list below) for 1 h in 5% milk on a rocking platform, then washed 3 times in PBS-T, then incubated for one hour with Horseradish Peroxidase (HRP)-conjugate secondary antibody (AffiniPure Goat anti-rabbit (1:5000; #111-035-003) and/or AffiniPure Goat anti-Mouse (1:10,000; #115-035-174) (Jackson ImmunoResearch)) and Streptactine HRP-conjugate (#1610380, Bio-Rad) for ladder staining. The membranes were then washed 3 times with PBS-T and one time with PBS before to be developed using Amersham™ ECL™ Prime Western Blotting Detection Reagent (GE Healthcare) and ChemiDoc system (Bio-Rad).

In *CCZ1* knockdown of PHH, the intensity of the bands corresponding to CCZ1 protein in the western blot were measured using ImageLab software (BioRad).

All uncropped blots are presented in Source data linked to this paper.

## Primary antibodies used for western blot

Rabbit anti-Human CCZ1 antibody (1:500) (#HPA050006, Sigma-Aldrich), Mouse monoclonal anti-human Rab7 antibody (1:2000) (#R8779, Sigma-Aldrich), Rabbit anti-Human NPC1 antibody (1:1000) (#SAB3500300, Sigma-Aldrich), Mouse monoclonal anti-human beta-actin antibody (1:10,000) (clone 8H10D10, #MA5-15452, Invitrogen), Mouse monoclonal anti-VSV-M antibody (1:500) (clone 23H12, #EB0011, Kerafast), Rabbit anti-Calnexin antibody (Home-made, 1:10,000).

## Statistical analysis

All analysis were done using the data from at least three independent experiments and are shown as ±SD in GraphPad Prism (Version 8.3.1). One-way ANOVA analyses and two-tailed student t-test were used as indicated in figure legends. *$P < 0.05$, **$P < 0.01$, ***$P < 0.001$; ****$P < 0.0001$. All exact $p$ values can be found in Source data linked to this paper.

## Data availability

The data generated in this study are provided in the Supplementary Information/Source Data file. Sequencing data are available on NCBI Sequence Read Archive under the accession number PRJNA1026654. Source data are provided with this paper.

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

## Acknowledgements

We would like to thank Prof. Manfred Boehm (NIH) for providing NC8 cells for blood-vessel organoids development. The work is supported by the Swedish research Council 2018-05766 (A.M., F.W. and J.M.P) and 2019-01837 (V.M.L) as well as the Bundesministerium für Bildung und Forschung (Infect-ERA, grant "ESCential") and the Pandemie Netzwerk Land Hessen (F.W.). This work is also supported by the Intramural Research Program of the NIAID, NIH (H.F.) and by the Vienna Science and Technology Fund (WWTF) through project COV20-002 (J.M.P), the Austrian Academy of Sciences and the City of Vienna and grants from the Austrian Science Fund (FWF) Wittgenstein award (Z 271-B19)(J.M.P), the T. Von Zastrow Foundation (J.M.P), the Innovative Medicines Initiative 2 Joint Undertaking (JU) under grant agreement No 101005026 (A.M., F.W., J.M.P), the Canada 150 Research Chairs Program F18-01336 (J.M.P), the Fundacio La Marato de TV3 (202125-31) (J.M.P) and the Canadian Institutes of Health Research COVID-19 grants F20–02343 and F20-

02015 (J.M.P.). This Joint Undertaking receives support from the European Union's Horizon 2020 research and innovation programme and EFPIA. We thank the proteomics core facility at IMBA for mass spectrometry analysis.

## Author contributions

This study was conceived and designed by V.M., A.M., C.S. and J.M.P. V.M., C.S. and L.J. wrote the manuscript. V.M., H.K. L.J., C.S., S.A. and S.D. performed all the experiments involving viruses/western blot/qRT-PCR/confocal microscopy/analysis. G.J., A.L. and A.H. developed blood-vessel organoids. S.U.V and S.Y. developed liver spheroids. S.Y. and V.M. run blood vessels confocal microscopy. M.D. and G.M. prepared the KO Vero E6 cells. A.L., N.L., J.V., U.E., M.F.S. and J.M.P developed the gene-trap haploid stem cells library screening used in this study and prepared the KO A549 cells. M.J.K. participated in data analysis and data visualization. M.D., G.M. and M.H. developed and analyzed the data from the mutagenesis haploid screening. H.F. developed the recombinant VSV-GP viruses. F.W., M.H. M.J.K. and V.M.L. helped with manuscript editing. A.M. and J.M.P. edited the manuscript.

## Funding

## Competing interests

H.F. claims intellectual property of VSV-based filovirus vaccines. All other authors declare no conflicts of interest.

## Additional information

[1]Karolinska Institute and Karolinska University Hospital, Department of Laboratory Medicine, Unit of Clinical Microbiology, Stockholm, Sweden. [2]National Veterinary Institute, Uppsala, Sweden. [3]Department of Molecular Medicine, University of Padova, Padova, Italy. [4]IMBA, Institute of Molecular Biotechnology of the Austrian Academy of Science, Vienna, Austria. [5]Vienna BioCenter PhD Program, Doctoral School of the University of Vienna and Medical University of Vienna, A-1030 Vienna, Austria. [6]Department of Physiology and Pharmacology, Karolinska Institutet, Stockholm, Sweden. [7]Public Health Agency of Sweden, Solna, Sweden. [8]Acus Laboratories GmbH, Cologne, Germany. [9]JLP Health GmbH, Vienna, Austria. [10]Institute for Virology, FB10-Veterinary Medicine, Justus Liebig University, Giessen, Germany. [11]University Tübingen, Tübingen, Germany. [12]Dr. Margarete Fischer-Bosch Institute of Clinical Pharmacology, Stuttgart, Germany. [13]Laboratory of Virology, National Institute of Allergy and Infectious Diseases, National Institutes of Health, Hamilton, MT, USA. [14]Department of Medical Genetics, Life Sciences Institute, University of British Columbia, Vancouver, BC, Canada. [15]Department of Laboratory Medicine, Medical University of Vienna, Vienna, Austria. [16]Helmholtz Centre for Infection Research, Braunschweig, Germany. [17]These authors contributed equally: Hyesoo Kwon, Lijo John, Cristiano Salata. ✉e-mail: ali.mirazimi@ki.se

