## [Peer Review File · Nature Communications]

REVIEWER COMMENTS

Reviewer #1 (Remarks to the Author):

In this manuscript, Monteil et al describe work that identifies CCZ1 as an important host factor for Marburg virus infection. The authors performed a screen in a haploid murine stem cell line infected with VSV Δ G/MARV GP and identify CCZ1 and RAB7 as significant hits. To validate CCZ1 as an important factor for MARV infection, they generated knockout (KO) clones of CCZ1, NPC1, and RAB7 in A549 cells using CRISPR/Cas9 and infected a mixture of KO and WT cells with VSV Δ G/MARV GP and MARV (and EBOV). This is followed by similar experiments in primary human hepatocyte culture and human blood-vessel organoids where CCZ1 was knocked down (KD) by siRNA or KO by CRISPR/Cas9 and MARV infection was decreased when CCZ1 was KD or KO. Additional immunofluorescence experiments were performed to show that CCZ1 affects endosomal trafficking and viral infection. Finally, the authors extend their observations to SARS-CoV-2. Overall, the authors present observations that can be interesting, but in order for this study to have an impact additional description and experiments are needed. The conclusions presented are not well supported by the data and, in many instances, the significance of their observations are overstated. Several figures lack descriptions, which makes them difficult to understand, and presented data lack appropriate controls. Moreover, the analysis of data relative to mock are presented in place of the actual measured data, which are not shown. This is problematic when assessing the quality of the data and experimental rigor.

Major comments:

The authors present in Fig. 2 the analysis of results from the haploid cell screen. However, missing in the corresponding text and figure legend are a description of the statistical tests that were performed. These should be included in this manuscript in addition to reference to citations. It would also be useful to identify the colored groups by function in the figure legend. If CCZ1 exists as a complex with MON1, why wasn't MON1 a hit in the initial screen and why did the authors not generate a MON1 KO clone for subsequent validation? These are critical assays to support the overall conclusions on the significance of CCZ1 in MARV infections.

Validation of the major gene hits is missing a direct measure of transcript removal. The western blots shown in Fig. 3 report detection of VSV-M in various KO clones upon infection with VSV Δ G/MARVGP. The western blots that are needed are direct detection of CCZ1, RAB7, and NPC1. Without this, there is not sufficient data to support that the genes are completely knocked out. With the westerns shown in Fig. 3, it is not clear whether the genes are essential for MARV infection as deletions of CCZ1, RAB7, and NPC1 do not prevent infection but show a decrease.

The authors try to show that CCZ1, RAB1, and NPC1 are important for VSV Δ G/MARVGP infection by mixing 70% of WT GFP A549 cells with 30% of GFP-negative KO cells and measuring survival to validate their results in Fig. 3d. It is not clear what the rationale for these experiments are. An appropriate experiment is to perform transcomplementation with the respective cDNAs and show rescue of infection.

Fig. 4a is missing a description of what data was collected, how the level of MARV infection was quantitated, and how the data was analyzed to generate the graph in Fig. 4a.

Fig. 4b, What are "control cells" (line 161)? Also, the statistical analysis between WT and CCZ1 KO clone 2 for MARV is labeled "ns" but is at similar levels as EBOV which has a $P < 0.05$. This is confusing.

Western blot shows that CCZ1 siRNA knockdown is incomplete in primary human hepatocytes (Fig. 5a) and it is not clear what data is represented in Fig. 5b, which shows small differences between control siRNA and CCZ1 siRNA.

Fig. 6 is lacking validation. The figure is missing data demonstrating that CCZ1 expression is knocked out in NC8 cells and that transcomplementation rescues infection. What data are being shown in Fig. 6c and 6d is not clear. Why was EBOV used to infect 3D-blood vessel organoids if the point is to validate CCZ1 as an important factor for MARV identified by their haploid screen?

It appears that CCZ1 plays a less critical role in MARV and EBOV infections of primary cells. CCZ1 is involved in a complex with other factors during early endosome trafficking, are these observations a general result of trafficking disruption?

Fig. 7 Very high MOI 20 is used. Control markers for different cellular structures are absent (lysosome, endosome). Colors in Fig. 7C are not identified.

Rationale and relevance for including SARS-CoV-2 in this study are lacking. Data shown in Fig. 8 contain similar problems as the other figures and reflect little impact on SARS-CoV-2 infection.

Additional comments:

There is a general lack of copyediting throughout the manuscript that leads to confusion. For example, Fig 1b. left panel image is very dark and difficult to see. Scale bars are missing from all of the images; stating magnification is not sufficient. Line 158, what is meant by "inactivation of CCZ1 and RAB7"? Fig. S1 references qRT-PCR data in Fig. 5E but there is no Fig. 5E in the current figures. Line 239, what is meant by "in adequacy..."?

Reviewer #2 (Remarks to the Author):

This study finds CCZ1 as an essential lysosomal trafficking regulator in Marburg and Ebola virus infections using human hepatocyte primary cells and human blood vessel organoids. Authors first performed genome-wide screening using mouse haploid embryonic stem cells (mESCs), and CCZ1 was identified as a potential mediator of virus infection by deep-sequencing in survived mESCs after infection with VSV-GP virus (a replication competent vesicular stomatitis virus pseudotyped with the Marburg virus glycoprotein). The validation study was conducted by using CRISPR/Cas9 gene editing and/or siRNA against CCZ1 in human diploid A549 cells, primary human hepatocyte, and human blood-vessel organoids. Overall, the experimental approaches in this study are innovative, and the significance of the major finding is high. However, the writing and data presentation can be improved with revisions. And one concern is that the effects of CCZ1 knock-out/inhibition in vivo is unclear. Followings are my suggestions for revisions.

1. The data presentation and main texts could be clearer for general readers to better understand the rationale, methodologies, results, and interpretation, if authors clarify key information in the main texts and figure legends. For example, in lines 111-115 and 122-126, the methodology is briefly explained, but it is not so clear what kind of materials are used and why these approaches are selected. I suppose authors meant that the mES cells are mixture of cells that have mutations in all genes, but I believe not all readers can understand this by just "genome-wide screening". Methods also do not explain the detail of this cell line. Figure 1 could be revised to clarify the methodology? The survived cells are meant to have mutations in genes that mediate virus cytotoxicity?
2. Some figure legends are too concise to understand what data are presented. For example, figure 4b says "Effect of the KO of CCZ1", but it is unclear what kind of data are presented in the figure. I suppose this is qPCR against the MARV virus sequencing and the values are normalized against mock virus infected cells, but this is not clear. The same is true to figure 5b, 6, and 8.
3. The majority of data rely on viral-specific qRT-PCR which is mean to show relative changes, and it is unclear what percentage of cells are infected. Figure S1 nicely show EBOV infection in green color, additional experiments will be helpful to clarify the impact of CCZ1 knock-out by showing the percentage of infected cells using flow cytometry and/or immunostaining.

4. qPCR shows statistically significant reduction of viral infection in in vitro models, yet the it is unclear if this level of reduction by CCZ1 knock-out/down is significant in the in vivo settings. This might be beyond the scope of this manuscript, but addition of this information would strengthen the significance of this study.
5. Figure 8a is missing the Y label.
6. Whole images of gel/blot should be provided for 5a, 8b.

Reviewer #3 (Remarks to the Author):

The manuscript "Identification of CCZ1 as an essential lysosomal trafficking regulator in Marburg and Ebola virus infections" by Monteil et al. describes a haploid cell screen for entry factors of filoviruses and the characterization of one of the top hits, CCZ1. Main findings are that CCZ1 is an essential factor for filoviruses acting at the level of early to late endosomal trafficking, and that this factor is also important for other viruses using this pathway for entry. As such, the manuscript is of relevance to a broad readership. Importantly, while the initial screen was done using the VSVdG/GP platform, which one could criticize as potentially problematic for entry studies, results were validated using infectious EBOV and MARV. Further, experiments were confirmed in highly relevant cell culture systems such as primary human hepatocytes and human blood-vessel organoids. From an experimental point of view, experiments are carefully designed and executed, and conclusions are justified by the data. Further, the manuscript is well written and easy to follow.

As such, I have only minor suggestions to further improve the paper:

Minor points:

- 1) line 301: A word seems to be missing in this sentence.
- 2) The use of "fold change over mock" to report RT-qPCR data was a bit confusing to me. Would it be possible to express these data more directly? I realize that the use of genome copies is most likely impossible, as data reflect results from a 1-step RT-qPCR from whole cell RNA extracts, but maybe the authors can come up with a more intuitive way to present these data?
- 3) Fig. 7C: While the quantification is convincing, the depicted microscopy pictures currently aren't. It might be helpful to show individual channels to make a stronger point for colocalization.
- 4) Is there a way to pharmaceutically target CCZ1? This should be discussed.

Answers to reviewers

Reviewer #1 (Remarks to the Author):

NCOMMS-23-05052

In this manuscript, Monteil et al describe work that identifies CCZ1 as an important host factor for Marburg virus infection. The authors performed a screen in a haploid murine stem cell line infected with VSV Δ G/MARV GP and identify CCZ1 and RAB7 as significant hits. To validate CCZ1 as an important factor for MARV infection, they generated knockout (KO) clones of CCZ1, NPC1, and RAB7 in A549 cells using CRISPR/Cas9 and infected a mixture of KO and WT cells with VSV Δ G/MARV GP and MARV (and EBOV). This is followed by similar experiments in primary human hepatocyte culture and human blood-vessel organoids where CCZ1 was knocked down (KD) by siRNA or KO by CRISPR/Cas9 and MARV infection was decreased when CCZ1 was KD or KO. Additional immunofluorescence experiments were performed to show that CCZ1 affects endosomal trafficking and viral infection. Finally, the authors extend their observations to SARS-CoV-2. Overall, the authors present observations that can be interesting, but in order for this study to have an impact additional description and experiments are needed. The conclusions presented are not well supported by the data and, in many instances, the significance of their observations are overstated. Several figures lack descriptions, which makes them difficult to understand, and presented data lack appropriate controls. Moreover, the analysis of data relative to mock are presented in place of the actual measured data, which are not shown. This is problematic when assessing the quality of the data and experimental rigor.

Response; We value these overall comments. In the updated manuscript, we have taken into account all of these concerns.

Major comments:

The authors present in Fig. 2 the analysis of results from the haploid cell screen. However, missing in the corresponding text and figure legend are a description of the statistical tests that were performed. These should be included in this manuscript in addition to reference to citations. It would also be useful to identify the colored groups by function in the figure legend.

Response: Thanks for the comment. It has been corrected/completed in figure 2, supplementary figure 1 and in the method (page 17 line 452-453)

If CCZ1 exists as a complex with MON1, why wasn't MON1 a hit in the initial screen and why did the authors not generate a MON1 KO clone for subsequent validation? These are critical assays to support the overall conclusions on the significance of CCZ1 in MARV infections.

Response: We appreciate the reviewers insightful comment. Similar to other screening platforms, our screening platform does not identify all essential genes. As mentioned by

reviewer, MON1, which consists of two proteins (MON1A and MON1B), form a complex with CCZ1 and other proteins.

In the revised manuscript, we have conducted experiments using MON1A and MON1B knockout cells to investigate the role of these proteins in infection. As depicted in figures 4f and g, the knockout of MON1A/B resulted in a reduction in the infection level, thus confirming the involvement of CCZ1 as an essential cellular factor within this complex in filovirus endocytosis.

Validation of the major gene hits is missing a direct measure of transcript removal. The western blots shown in Fig. 3 report detection of VSV-M in various KO clones upon infection with VSVΔG/MARVGP. The western blots that are needed are direct detection of CCZ1, RAB7, and NPC1. Without this, there is not sufficient data to support that the genes are completely knocked out. With the westerns shown in Fig. 3, it is not clear whether the genes are essential for MARV infection as deletions of CCZ1, RAB7, and NPC1 do not prevent infection but show a decrease.

Response: In revised manuscript, we add the data which is highlighted by the reviewer (western-blot showing the KO expression of CCZ1, RAB7, and NPC1) as supplementary figure 2.

The authors try to show that CCZ1, RAB1, and NPC1 are important for VSVΔG/MARVGP infection by mixing 70% of WT GFP A549 cells with 30% of GFP-negative KO cells and measuring survival to validate their results in Fig. 3d. It is not clear what the rationale for these experiments are. An appropriate experiment to perform is transcomplementation with the respective cDNAs and show rescue of infection.

The objective of this experiment was to validate the impact of CCZ1 knockout under identical conditions to the wild-type cells. This involved placing all the cells in the same well and infecting them simultaneously with an equal amount of virus to minimize potential variations.

However, in revised Manuscript we add transcomplementation experiments for CCZ1 and MON1 in A549. We didn't transcomplement for NPC1 and RAB7 as other studies has already showed their role in filovirus infection.

Anyway, using reversible mutant haploid cells furnished by Haplobank (IMBA) (Elling U. *et al*, Nature, 2017), we were able to show that the knockout of *ccz1*, *npc1* and *rab7* reduce the level of infection and that reversion of the knockout for all three genes reverse the effect of the knockout on infection. These data are now presented as figure 3. Data on transcomplementation of CCZ1, MON1A and MON1B in A549 are shown in figure 4 e-g.

Fig. 4a is missing a description of what data was collected, how the level of MARV infection was quantitated, and how the data was analyzed to generate the graph in Fig. 4a.

Response: In the revised manuscript, the figure is now figure 5a and we have included details in the legend as well as in the methods section to ensure clarity (page 19 Line 487-

490)

Fig. 4b, What are “control cells” (line 161)? Also, the statistical analysis between WT and CCZ1 KO clone 2 for MARV is labeled “ns” but is at similar levels as EBOV which has a $P < 0.05$. This is confusing.

Response; Thank you for this remark. Control cells are cells submitted to CRISPR/Cas9 treatment using a scrambled RNA guide. This information has been added in line 211 in revised MS.

For MARV, the variability between the samples is higher than for EBOV, that might explain the “non-significant. In order to make the figure clearer, all points are now showed in all the figures of the manuscript.

Western blot shows that CCZ1 siRNA knockdown is incomplete in primary human hepatocytes (Fig. 5a) and it is not clear what data is represented in Fig. 5b, which shows small differences between control siRNA and CCZ1 siRNA.

Response: The data presented in Figure 5b has been further clarified in the figure legend. We acknowledge that the siRNA knockdown may not be complete, but it was still effective in significantly reducing the infection level in PHH. Additionally, the scale used in the other graph is logarithmic. To enhance the visualization of the reduction in virus infection, we modified the data presentation to show the infection level relative to the control siRNA and added the percentage of infection decrease in the text (line 228-229)

Fig. 6 is lacking validation. The figure is missing data demonstrating that CCZ1 expression is knocked out in NC8 cells and that transcomplementation rescues infection. What data are being shown in Fig. 6c and 6d is not clear. Why was EBOV used to infect 3D-blood vessel organoids if the point is to validate CCZ1 as an important factor for MARV identified by their haploid screen?

Response; The relative *CCZ1* expression (mRNA level) is shown in figure 7b (previously 6b). To show the protein level, the cells were submitted to mass spectrometry analysis. The data were normalized to a housekeeping protein. The data are now also shown in figure 7b. All data are now better explained in the legend. We have already demonstrated the specificity of CCZ1 by using trancomplementations experiment in A549 in revised MS (Figures 3 and 4e-g)

We also add Ebola virus side by side to Marburg virus as Ebola is the another member of the Filoviridae family with public health importance.

It appears that CCZ1 plays a less critical role in MARV and EBOV infections of primary cells. CCZ1 is involved in a complex with other factors during early endosome trafficking, are these observations a general result of trafficking disruption?

Response; : This is a good comments, however, we believe that the shut down of genes in primary cells in this case PHH, is much more complicated and thereby the effect on infection is limited. However, this has been discussed in revised MS (line 353-357)

Fig. 7 Very high MOI 20 is used. Control markers for different cellular structures are absent (lysosome, endosome). Colors in Fig. 7C are not identified.

Response: In order to visualize the entering viruses inside vesicles in the cells, we utilized a high MOI as the sensitivity of detecting these particles through IFA is limited. We believe that including additional markers for cellular structures in Figure 7b would result in visual clutter without providing any additional information. However, in Figure 7c, we successfully demonstrated through colocalization studies that the virus is indeed trapped in the early endosome when ccz1 is knocked out. Colors are now identified in the legend of figure 7c.

Rationale and relevance for including SARS-CoV-2 in this study are lacking. Data shown in Fig. 8 contain similar problems as the other figures and reflect little impact on SARS-CoV-2 infection.

Response: A screening using SARS-CoV-2 was run and ccz1 was one of the hit. It's now explained (line 289-296) and data from the screening were added as supplementary figure 6.

The scale of the figure is logarithmic and doesn't properly show the effect of the KO of CCZ1 on SARS-2 infection. We now changed the presentation of the data to ameliorate the visualization of the decrease in infection, and added the values in the text (line 306 and 308).

Additional comments:

There is a general lack of copyediting throughout the manuscript that leads to confusion. For example, Fig 1b. left panel image is very dark and difficult to see. Scale bars are missing from all of the images; stating magnification is not sufficient. Line 158, what is meant by "inactivation of CCZ1 and RAB7"? Fig. S1 references qRT-PCR data in Fig. 5E but there is no Fig. 5E in the current figures. Line 239, what is meant by "in adequacy...?"

Response: Thank you for these comments. It is now corrected (Scale bars on all the images; Inactivation page 8 line 205, FigS1 became figure S3 and the reference was modified; In adequacy became in agreement page 1 line 302)

Reviewer #2 (Remarks to the Author):

This study finds CCZ1 as an essential lysosomal trafficking regulator in Marburg and Ebola virus infections using human hepatocyte primary cells and human blood vessel organoids. Authors first performed genome-wide screening using mouse haploid embryonic stem cells (mESCs), and CCZ1 was identified as a potential mediator of virus infection by deep-sequencing in survived mESCs after infection with VSV-GP virus (a replication competent vesicular stomatitis virus pseudotyped with the Marburg virus glycoprotein). The validation study was conducted by using CRISPR/Cas9 gene editing and/or siRNA against CCZ1 in human diploid A549 cells, primary human hepatocyte, and human blood-vessel organoids.

Overall, the experimental approaches in this study are innovative, and the significance of the major finding is high. However, the writing and data presentation can be improved with revisions. And one concern is that the effects of CCZ1 knock-out/inhibition in vivo is unclear. Followings are my suggestions for revisions.

Response; We value these overall comments. In the updated manuscript, we have taken into account these advice by the reviewer.

1. The data presentation and main texts could be clearer for general readers to better understand the rationale, methodologies, results, and interpretation, if authors clarify key information in the main texts and figure legends. For example, in lines 111-115 and 122-126, the methodology is briefly explained, but it is not so clear what kind of materials are used and why these approaches are selected. I suppose authors meant that the mES cells are mixture of cells that have mutations in all genes, but I believe not all readers can understand this by just “genome-wide screening”. Methods also do not explain the detail of this cell line. Figure 1 could be revised to clarify the methodology? The survived cells are meant to have mutations in genes that mediate virus cytotoxicity?

Response. Thank for your comment. It is now modified to give more details (Figure 1 as well as page 4 line 113-122)

2. Some figure legends are too concise to understand what data are presented. For example, figure 4b says “Effect of the KO of CCZ1”, but it is unclear what kind of data are presented in the figure. I suppose this is qPCR against the MARV virus sequencing and the values are normalized against mock virus infected cells, but this is not clear. The same is true to figure 5b, 6, and 8.

Response: Thank you for good remark. The legends for all the figures are now more detailed to make the data understandable.

3. The majority of data rely on viral-specific qRT-PCR which is mean to show relative changes, and it is unclear what percentage of cells are infected. Figure S1 nicely show EBOV infection in green color, additional experiments will be helpful to clarify the impact of CCZ1 knock-out by showing the percentage of infected cells using flow cytometry and/or immunostaining.

Thank you for this remark. In revised manuscript we have add additional data demonstrating the infection rate by IF pictures (Figure 5b and supplementary figure 3)

4. qPCR shows statistically significant reduction of viral infection in in vitro models, yet the it is unclear if this level of reduction by CCZ1 knock-out/down is significant in the in vivo settings. This might be beyond the scope of this manuscript, but addition of this information would strengthen the significance of this study.

Response: Thank you for your comment. We fully agree that an *in vivo* experiment would strengthen the significance of the data. Unfortunately, to our knowledge, there is no animal model KO for *ccz1*, particularly mice that is the only animal we can handle in our BSL4. However, this is beyond the scope of this report.

5. Figure 8a is missing the Y label.

Response: Thank you for this very good point. It is now corrected (figure 9a).

6. Whole images of gel/blot should be provided for 5a, 8b.

Response; Thank you. The whole blot are now furnished in the source data linked to this paper.

Reviewer #3 (Remarks to the Author):

The manuscript “Identification of CCZ1 as an essential lysosomal trafficking regulator in Marburg and Ebola virus infections” by Monteil et al. describes a haploid cell screen for entry factors of filoviruses and the characterization of one of the top hits, CCZ1. Main findings are that CCZ1 is an essential factor for filoviruses acting at the level of early to late endosomal trafficking, and that this factor is also important for other viruses using this pathway for entry. As such, the manuscript is of relevance to a broad readership. Importantly, while the initial screen was done using the VSVdG/GP platform, which one could criticize as potentially problematic for entry studies, results were validated using infectious EBOV and MARV. Further, experiments were confirmed in highly relevant cell culture systems such as primary human hepatocytes and human blood-vessel organoids. From an experimental point of view, experiments are carefully designed and executed, and conclusions are justified by the data. Further, the manuscript is well written and easy to follow.

Response; We value these overall comments.

As such, I have only minor suggestions to further improve the paper:

Minor points:

1) line 301: A word seems to be missing in this sentence.

Response: Thank you. This has been corrected in revised MS.

2) The use of “fold change over mock” to report RT-qPCR data was a bit confusing to me. Would it be possible to express these data more directly? I realize that the use of genome copies is most likely impossible, as data reflect results from a 1-step RT-qPCR from whole

cell RNA extracts, but maybe the authors can come up with a more intuitive way to present these data?

Response: Thank you for this comment. In revised MS, we also included the percentage of infection decrease in the text to make the data more clear, changed the wording to present the data and added explanation in the method.

3) Fig. 7C: While the quantification is convincing, the depicted microscopy pictures currently aren't. It might be helpful to show individual channels to make a stronger point for colocalization.

Response: Thank you for good comment. The individual channels are now shown in supplementary figure 5.

4) Is there a way to pharmaceutically target CCZ1? This should be discussed.

Response: This is interesting comment. To our knowledge, there is no inhibitor developed for ccz1 but there is an open-field for it. It is now better discussed (page 15 line 381-384)

REVIEWERS' COMMENTS

Reviewer #1 (Remarks to the Author):

In this revised manuscript, Monteil and colleagues have adequately addressed most concerns raised by the reviewers to describe a haploid murine cell screen to identify factors important for filoviral infection and the validation of CCZ1. These changes have improved the overall manuscript.

Reviewer #2 (Remarks to the Author):

Authors nicely responded to my comments. However, the quantitative result in supplementary figure 4 seems missing, while the main texts (lines 242-245) describe the reduction percentage of infection. Please include the quantitative results in the figure and update the figure legend.

Reviewer #3 (Remarks to the Author):

The authors have adequately addressed my comments.

Answers to reviewers

Reviewers comments:

Remaining reviewer comments:

Reviewer #1:

In this revised manuscript, Monteil and colleagues have adequately addressed most concerns raised by the reviewers to describe a haploid murine cell screen to identify factors important for filoviral infection and the validation of CCZ1. These changes have improved the overall manuscript.

Response: Thank you

Reviewer #2:

Authors nicely responded to my comments. However, the quantitative result in supplementary figure 4 seems missing, while the main texts (lines 242-245) describe the reduction percentage of infection. Please include the quantitative results in the figure and update the figure legend.

Response: Thank you. The percentage used in the text being calculated using data from figure 7 and not IF data from supp fig 4, values have been added to figure 7.

Reviewer #3:

The authors have adequately addressed my comments

Response: Thank you